# A CD8+ NK cell transcriptomic signature associated with clinical outcome in relapsing remitting multiple sclerosis

Eoin F. McKinney [1,2 ✉], Iona Cuthbertson [2], Kristina M. Harris [3], Dawn E. Smilek[3], Christopher Connor [2], Giulia Manferrari[2], Edward J. Carr [2], Scott S. Zamvil [4] & Kenneth G. C. Smith [1,2]

Multiple sclerosis (MS) is an inflammatory demyelinating disease of the central nervous system (CNS) with the majority of cases characterised by relapsing/remitting (RRMS) attacks of neurologic dysfunction followed by variable resolution. Improving clinical outcomes in RRMS requires both a better understanding of the immunological mechanisms driving recurrent demyelination and better means of predicting future disease course to facilitate early targeted therapy. Here, we apply hypothesis-generating network transcriptomics to CD8+ cells isolated from patients in RRMS, identifying a signature reflecting expansion of a subset of CD8+ natural killer cells (NK8+) associated with favourable outcome. NK8+ are capable of regulating CD4+ T cell activation and proliferation in vitro, with reduced expression of HLA-G binding inhibitory receptors and consequent reduced sensitivity to HLA-G-mediated suppression. We identify surrogate markers of the NK8+ signature in peripheral blood leucocytes and validate their association with clinical outcome in an independent cohort, suggesting their measurement may facilitate early, targeted therapy in RRMS.

[1] Cambridge Institute for Therapeutic Immunology and Infectious Disease, Jeffrey Cheah Biomedical Centre, Cambridge, UK. [2] Department of Medicine, University of Cambridge School of Clinical Medicine, Cambridge, UK. [3] Biomarker Discovery Research, ITN, Bethesda, MD, USA. [4] Department of Neurology and Program in Immunology, University of California, San Francisco, CA, USA. ✉email: efm30@cam.ac.uk

The process of T cell exhaustion—by which CD8+ T cells acquire effector dysfunction in the face of persistent antigen stimulation—has been associated with clinical outcome, not only in multiple distinct autoinflammatory diseases but also in both chronic viral infections and anti-tumour immunity[1]. In relapsing remitting multiple sclerosis (RRMS), relapse resolution associates with favourable long-term outcome[2,3]; however, the frequency and manifestations of early attacks are highly variable and the immunopathology of RRMS onset, relapse and progression remains poorly understood[4]. Distinct immunological pathways are thought to drive relapse and secondary neurodegenerative progression[5] with CD4+ T cells remaining the prime suspect for initiating demyelination in both human disease and animal models[5]. Natural killer (NK) cells are an innate lymphocyte subset capable of eliminating cells identified as either missing-self or altered self[6]. Missing self−elimination occurs when downregulation of self major histocompatibility complexes (MHC, e.g., on virally infected cells[7]) attenuates signalling through NK inhibitory receptors. Altered self responses are driven by activating NK receptors whose ligands are upregulated on transformed, stressed or activated cells[7]. NK cells may also play an immunoregulatory role[8], as highlighted by the enhanced infection-induced immunopathology in NK-deficient mice, where NK cells fail to control autologous T cell activation[9]. Genetic variants or alterations in NK cell function or phenotype have been repeatedly linked to multiple autoimmune diseases including rheumatoid arthritis[10], Type 1 diabetes (T1D)[11] and RRMS[12], although it remains unclear whether this reflects a contribution to tissue injury or failure to control immune reactivity[13].

Here we apply unsupervised network transcriptomics to circulating CD8+ leucocytes isolated from patients who had experienced a first demyelinating episode (clinically isolated syndrome, CIS), were not on treatment and were at high risk for RRMS relapse in a phase II randomised, placebo-controlled trial of the immunodulatory effect of atorvastatin (STAyCIS)[14]. Samples were taken prior to commencing therapy in the trial to ensure this did not impact on the baseline result. In the STAyCIS study we observe modules of coexpressed transcripts in this CD8+ population that are associated with favourable clinical outcome, are independent of treatment effects and reflect expansion of a CD8+CD3−CD56+ natural killer cell subset (NK8+) rather than a profile of CD8+ T cell exhaustion. In vitro studies indicate that NK8+ can regulate autologous CD4+ T cell activation and proliferation with reduced sensitivity to human leucocyte antigen G (HLA-G) mediated suppression. We validate the association of an NK8+ gene signature with reduced relapse risk in peripheral blood mononuclear cells from an independent cohort of 94 RRMS and CIS cases.

## Results

**Network coexpression analysis of early MS.** Unsupervised analyses of high-throughput 'omic' data facilitate the discovery of previously unsuspected associations as they are, by definition, not founded on existing knowledge[15]. We performed an unsupervised transcriptomic analysis of CD8+ cells isolated from patients enrolled after a first demyelinating episode into a randomised, double-blind, placebo-controlled clinical trial (STAyCIS[14]) that tested the effect of atorvastatin on relapse in patients in the earliest clinical phase of MS. CD8+ cells were isolated from peripheral blood mononuclear cells (PBMC) collected at the baseline trial visit (Supplementary Fig. 1), allowing unsupervised identification of coexpressed gene 'modules' (weighted gene coexpression analysis, WGCNA[16]). 'Eigengenes' summarising modular coexpression signatures were then correlated to prospective and baseline clinical data, including the primary endpoint of the study (PEP, one clinical relapse or ≥3 new T2 MRI lesions within 12 months, Fig. 1A), the treatment arm (atorvastatin or placebo) and technical covariates (Supplementary Fig. 2 and Supplementary Data 1). Four CD8+ cell-intrinsic eigengenes were specifically and significantly correlated with clinical and radiological study endpoints (Fig. 1A, B). While transcripts associated with T cell exhaustion[1] were coexpressed in the RRMS dataset, they were not associated with clinical outcome (Supplementary Fig. 2E, F).

**Biological interpretation of progression-associated transcriptional signature.** Next, we sought to interpret gene modules associated with relapse risk by performing enrichment analysis against public repositories of immune signatures[17]. The largest module ('black', Supplementary Data 2) was strongly enriched only for NK-cell specific transcripts as defined in two distinct immune signature repositories[18,19], while smaller modules showed no clear enrichment (Fig. 1C). Supervised clustering using the 'black' transcripts identified patient subgroups with distinct outcomes that were comparably enriched for NK transcripts (Fig. 1D, E). This NK signature was not correlated with basline clinical traits (Supplementary Data 1) and was the strongest predictor identified in a multivariate penalised Cox regression model of time to relapse, although measures of clinical severity (T1 lesion load and clinical severity) further improved prediction when used alongside it (Fig. 1H). In humans, a subset of NK cells also express the CD8 coreceptor[20], albeit at lower levels than their T cell counterparts, and flow cytometry confirmed that CD3−CD56+CD8+ NK cells (NK8+) comprised a significant minority of the isolated CD8+ fraction profiled by transcriptomics (NK8+, Supplementary Fig. 3A, B). Correlation of the outcome-associated 'black' signature with extensive concurrent immunophenotyping confirmed robust, specific association with an NK8+ expansion in peripheral blood (Fig. 1F, Supplementary Fig. 3D–G). To further confirm NK8+ cells as the source of this signature, we isolated NK8+ and NK8− cells from healthy individuals, performing RNAseq to identify a signature defining the NK8+ subset, and demonstrated strong differential enrichment between disparate outcome groups identified with the 'black' signature (Fig. 1G and Supplementary Data 3). Together these data demonstrate that a transcriptional signature reflecting an expanded population of NK8+ cells is associated with reduced future relapse risk following an initial demyelinating event.

**An autoregulatory role for NK8+ cells.** We next sought to investigate potential immune mechanisms explaining the association of NK8+ cells with reduced relapse risk. Cytotoxic NK cell function is controlled by a complex interplay of both activating and inhibitory signals[21]. As an NK8+ expansion associated with reduced relapse risk, we hypothesised that NK8+ might play an immunoregulatory role, limiting recurrent T-cell driven demyelination in RRMS. To test this, we stimulated CD4+ or CD8+ T cells in the presence of a titrated ratio of autologous NK8+ or NK8− cells (Supplementary Fig. 4A), while taking care to prevent NK pre-activation (Supplementary Fig. 4B, C). We found that, compared to their NK8− counterparts, NK8+ cells exerted a significant suppressive effect on autologous CD4+ (Fig. 2A–D) but not CD8+ T cell (Fig. 2E–H) proliferation and activation, although we cannot exclude that higher NK:T cell ratios or reduced CD8 TCR stimulation levels may allow CD8 T cell suppression to also occur. As no differences were observed in the cytotoxic granule number or content of NK8+ and NK8− cells (by either flow cytometry or electron microscopy, Supplementary Fig. 5), we asked whether an altered balance of activating and

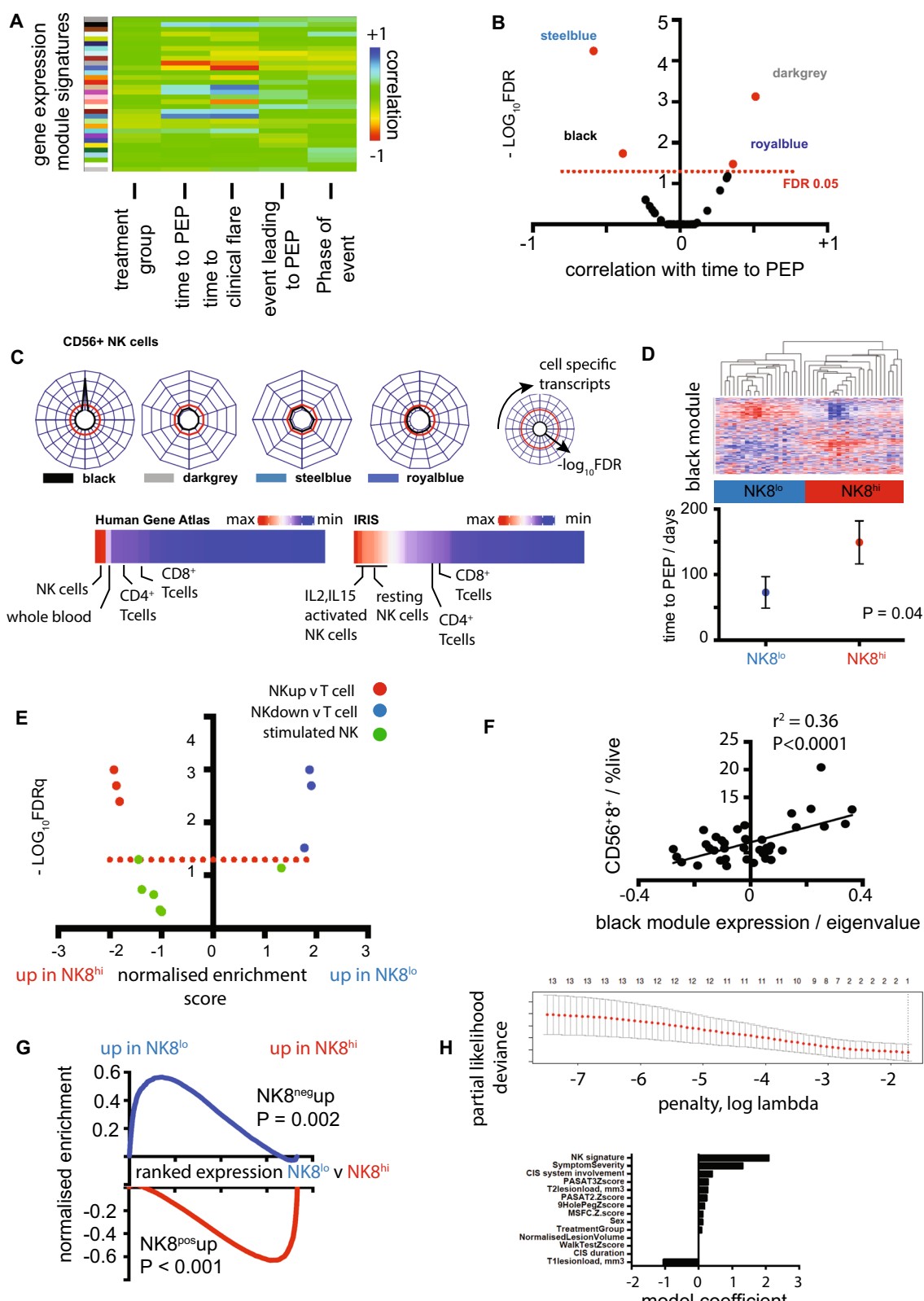

inhibitory signals might explain the superior ability of NK8+ cells to inhibit autologous CD4+ T cell activation. Using a validated flow cytometry panel[22] we quantified the expression of NK killer inhibitory and activating receptors (Supplementary Data 4) on NK8+, NK8− and CD56hi subsets in PBMC from healthy individuals. While most receptors were equivalently expressed (Supplementary Fig. 6), we observed increased expression on NK8+ of

the activating receptor NKG2D, and reduced expression of two inhibitory receptors (ILT2, KIR2DL4, Fig. 3A). While KIR2DL4 may serve as an activating or inhibitory receptor[23], both ILT2 and KIR2DL4 share the same ligand, HLA-G, a non-classical HLA molecule first identified as a key regulator of foetal–maternal tolerance[24] and now recognised to play a regulatory role in suppressing immunity during infection, transplantation,

**Fig. 1 An NK8 signature associates with relapse risk in multiple sclerosis. A** Heatmap showing correlation of gene expression eigengenes (coloured blocks, y-axis) with clinical endpoints of the STAyCIS trial (Pearson correlation, colour bar). Phase of event = event occurring during a 12 m treatment phase or extended follow-up to 18 m. Events leading to the primary endpoint (PEP) were classified as clinical or radiological. **B** Volcano plot showing modules with significant association with PEP (x = Pearson correlation of module eigengene and time to event, y = $-\log_{10}$FDR). **C** Radar plots showing enrichment ($-\log_{10}$FDR) of outcome-associated modular signatures (from panels **A**, **B**) compared to immune cell-specific signatures from the Human gene atlas (HGA, upper) and heatmaps showing 'black' module eigengene expression across all cell types in HGA and immune response in silico (IRIS) repositories. Each point on the circumference reflects a different set of cell-specific transcripts including all (of 79 total) showing any overlap with modular signatures. **D** Heatmap (upper panel, patients in columns, genes in rows red:blue =max:min expression) and dotplot (lower panel) showing unsupervised hierarchical clustering using the 'black' module (panel **A**) and associated time to PEP of identified patient subgroups. **e** Volcano plot showing enrichment of NK-associated signatures from the IRIS dataset in patient subgroups identified in (**D**). **F** Scatterplot illustrating correlation of the black module eigengene (x-axis) against %CD56$^+$CD8$^+$ cells. **G** Enrichment of an NK8$^+$-specific signature, obtained by flow sorting/RNAseq of CD8$^+$ v CD8$^-$ NK cells from five healthy individuals in CD8$^+$ leucocytes from STAyCIS participants with NK8hi/lo signature at baseline as illustrated in (**c**). **H** Scatterplot (top) and barplot (bottom) illustrating cross-validation and feature-specific weighted coefficients, respectively, in a penalised Cox proportional hazards model using a naive elastic net. Scatterplot shows cross-validated error (deviance $+/-$ sem, y-axis) against increasing regularisation penalty ($\log \lambda$, x-axis). PEP = primary endpoint, FDR = false discovery rate, sem = standard error of the mean, FDR threshold on radarplots = 0.05 (red dashed line). Phase of event = event occurring during 12 m treatment phase or extended follow-up to 18 m. CIS = clinically isolated syndrome, PASAT = Paced Auditory Serial Addition Test, MSFC = Multiple Sclerosis Functional Composite. Error bars = mean $+/-$ sem.

autoimmunity and cancer[25]. Polymorphic variation at the HLA-G locus dictates expression levels of the receptor and has been associated with altered susceptibility to multiple autoimmune diseases[26], including MS[27]. We observed that HLA-G was upregulated on CD4$^+$ T cell activation (Supplementary Fig. 7A) and hypothesised that NK8$^+$ may be relatively refractory to its suppressive effect by virtue of their reduced HLA-G receptor expression. To test this, we investigated the ability of HLA-G to differentially inhibit NK8$^+$ and NK8$^-$ responses in a cytotoxicity assay. Isolated NK8$^+$ or NK8$^-$ cells were co-cultured in the presence of target cell lines with cytotoxicity measured by surface translocation of the granule protein CD107[28]. In each case we observed similar levels of induced cytotoxicity (Supplementary Fig. 7B), but in the presence of a titrated dose range of soluble HLAG (sHLAG) we observed a reduction in cytotoxicity of NK8$^-$ but not NK8$^+$ cells (Fig. 3C). Thus, NK8$^+$ cells can play an immunoregulatory role, mediated through suppression of CD4$^+$ T cell activation and proliferation, and appear relatively impervious to the inhibitory effects of HLA-G through limited expression of HLA-G receptors.

NK8$^+$ cells may therefore be capable of regulating proliferation and activation of autologous CD4$^+$ T cells, through a mechanism at least partially dependent on reduced HLA-G-mediated NK suppression.

**Validating NK8 prediction of clinical outcome.** Having identified a potential mechanism for the NK8$^+$ gene signature associated with favourable outcome in STAyCIS, we sought to confirm the finding in an independent cohort of patients with RRMS and CIS. To facilitate this, we used a cohort of patients with anti-neutrophil cytoplasmic antibody-associated vasculitis (Supplementary Data 6) to first identify surrogate markers of the NK8$^+$ signature that could be measured in a mixed peripheral blood mononuclear cell (PBMC) population (Supplementary Fig. 8, Supplementary Data 5). to stratify a second prospective clinical RRMS study[29] (Supplementary Data 7) into subgroups enriched or depleted for NK8$^+$ transcripts (Fig. 3D, analogous to the STAyCIS subgroups shown in Fig. 1D). As observed in the STAyCIS trial, favourable clinical outcome (clinical relapse-free survival) was associated with enrichment of surrogate markers of the NK8$^+$ signature (Fig. 3D–F) in this dataset derived from PBMC. These data independently validate the finding that an NK8$^+$ expansion is associated with favourable prognosis, supporting its use as a predictive tool to guide early aggressive therapy in RRMS.

## Discussion

Our observations are consistent with and extend previous reports of NK-mediated control of autoimmunity in RRMS. Active RRMS is known to be characterised by periods of reduced NK cell number and cytotoxicity in both cerebrospinal fluid (CSF) and peripheral blood while expansion of immature CD56$^{bright}$ NK cells accompanied clinical response to immunotherapy[8,30,31]. The comparatively small population of immature CD56$^{hi}$ cells has a well-described immunoregulatory role, dependent on NKG2D-mediated regulation of CD4 T cell activation[32]. Our findings indicate that NK8$^+$ cells contribute to regulation of autologous CD4$^+$ T cells, and that HLA-G expression on activated CD4$^+$ T cells may allow escape from CD8$^-$ CD56$^{dim}$ NK-mediated regulation. Therapeutic strategies promoting NK8$^+$ differentiation or altered T cell HLA-G expression may have coordinated regulatory effects on pathogenic T cells to limit demyelination in CIS/RRMS.

While clinical translation will require further validation work, the data presented here suggest that measurement of an NK8$^+$ cell-associated transcriptional signature could potentially be used to facilitate prediction of clinical outcome to target therapy in RRMS, and guide early treatment decisions.

## Methods

**The STAyCIS trial and sample collection.** The STAyCIS trial[14] is a randomised, double-blind, placebo-controlled, multicentre study evaluating the efficacy and safety of atorvastatin (Lipitor, Pfizer, 80 mg/day) in patients with clinically isolated syndrome (a first demyelinating event) and at high risk of conversion to RRMS. Written informed consent was obtained from patients prior to enrolment in the STAyCIS study (NCT00094172). Eighty-two participants were recruited within a screening phase of 90 days from the index CIS event and followed up for 18 months (12 months treatment phase) with serial clinical and radiological (MRI imaging) review. The primary combined endpoint was the development of either radiological ($\geq$3 new T2 MRI lesions) or clinical relapse ($\geq$1 clinical exacerbation) during the 12-month treatment phase. The trial was sponsored by NIAID in collaboration with the Immune Tolerance Network (clinicaltrials.gov NCT00094172).

The study was approved by institutional review boards at 14 centres in the United States and Canada. Written informed consent was obtained from patients prior to enrolment in the STAyCIS study (NCT00094172).

**Sample Collection, processing and QC.** Fifty-six STAyCIS PBMC samples for which viable cells were available were isolated from peripheral blood by centrifugation over Histopaque (Invitrogen) before controlled freezing at 1 °C/min (with isopropyl alcohol, Nalgene) and storage in buffer (10%FBS/DMSO) in liquid nitrogen. Frozen samples were rapidly thawed in the presence of endonucleases (Benzonase, Merck) at 37 °C, stained with anti-CD8 microbeads (Miltenyi) and enriched using a MACS column (Miltenyi).

**Transcriptomic data generation and QC.** Aliquots of total RNA (200 ng) were labelled using Ambion WT sense Target labelling kit and hybridised to Human

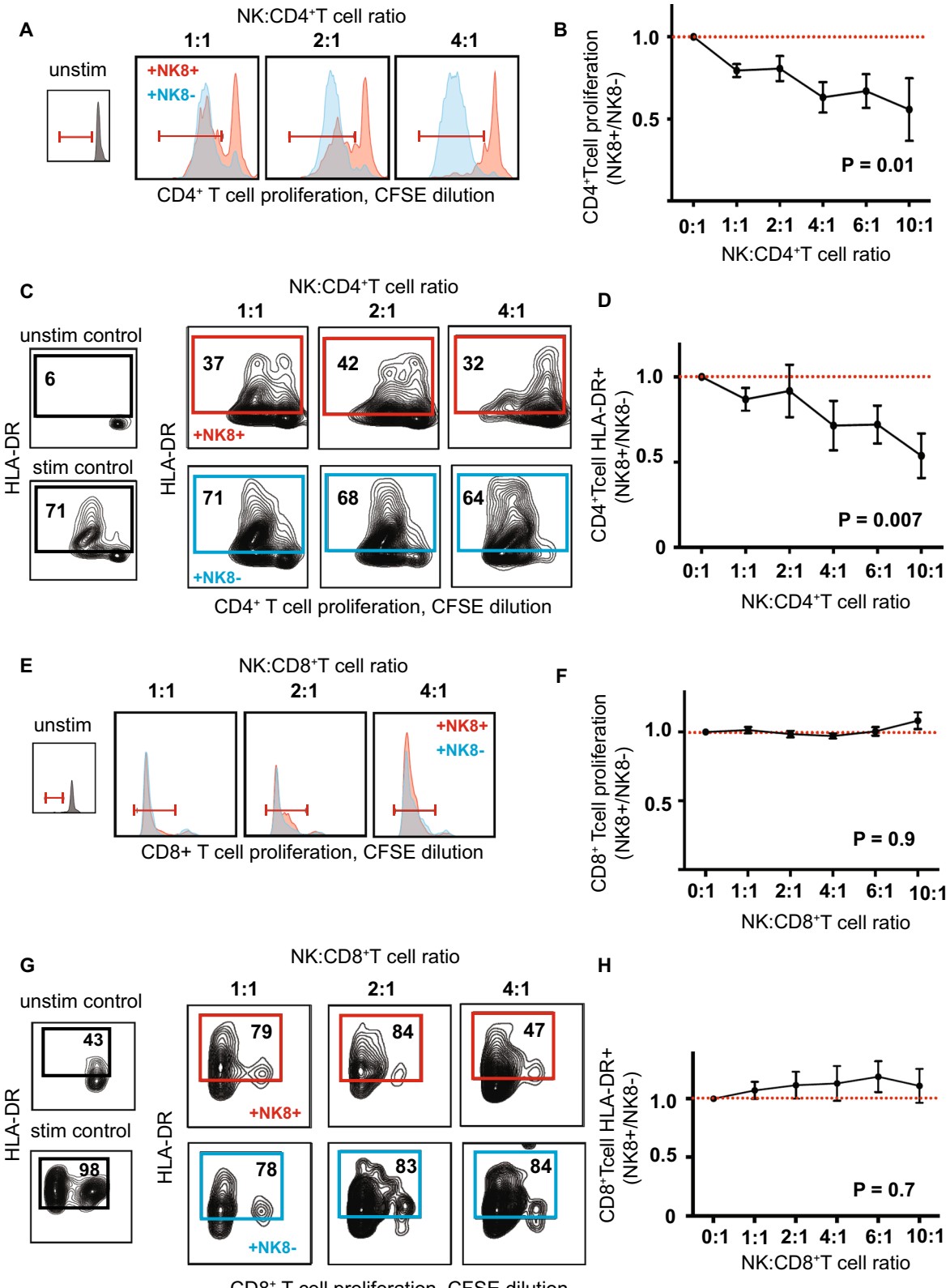

**Fig. 2 NK8 cells regulate autologous CD4+ T cell activation.** Representative histograms (**A**, **E**), contour plots (**C**, **G**) and summary scatter/line plots (**B**, **D**, **F**, **H**) showing flow cytometric quantification of CD4+ (**A–D**) and CD8+ (**E–H**) T cell proliferation (**A**, **B**, **E**, **F**, CFSE dilution) and activation (HLADR expression, **C**, **D**, **G**, **H**) when unstimulated (inset, black) or following polyclonal stimulation in vitro (anti-CD2/3/28 bead) in the presence of a titrated ratio of autologous NK8+ (red) or NK8− (blue) cells as indicated. Summary NK8+ results (**B**, **D**, **F**, **H**) expressed as ratio relative to paired NK8− cell-enriched culture. $P$ = 2-way ANOVA with NK:T ratio as the categorical variable. For b, d, f and h $n = 5$ independent biological replicates per group. Error bars = mean +/− sem, red dashed line = no difference vs control.

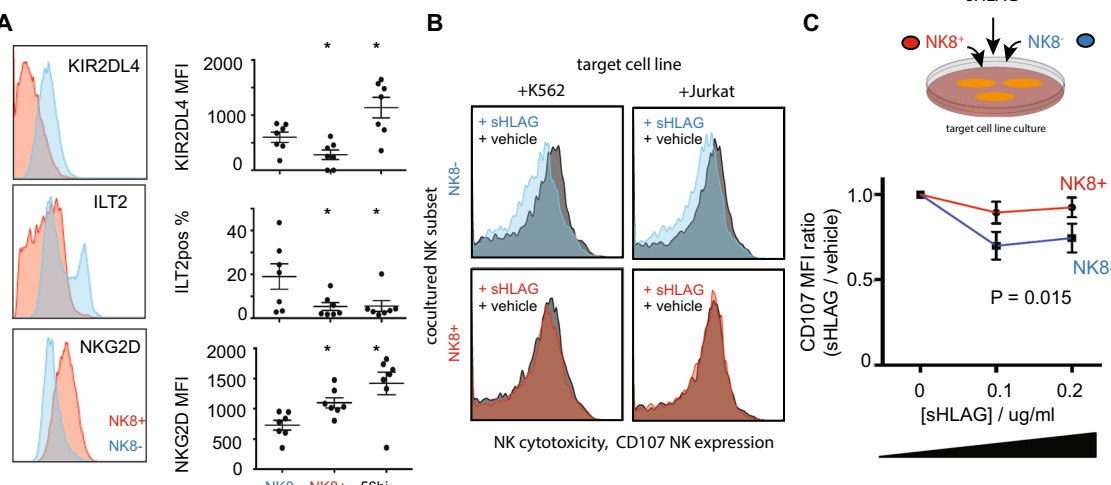

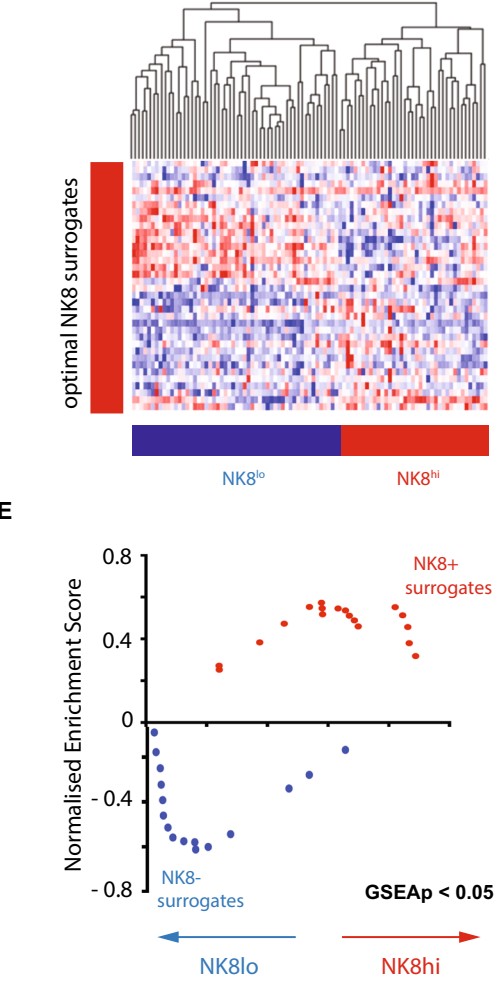

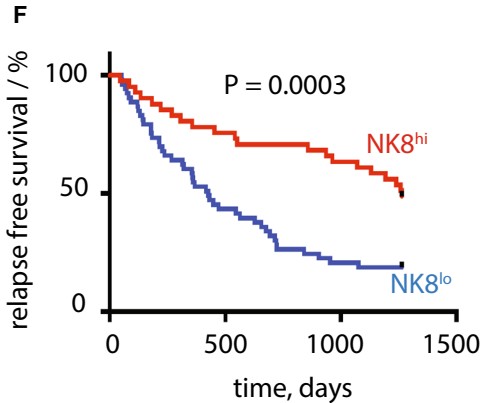

Gene 1.0 Arrays (Affymetrix) following the manufacturer's instructions. After washing, arrays were scanned using a GS 3000 scanner (Affymetrix) and CEL files were imported into RStudio (version 3.5.1) for QC and analysis. Affymetrix raw data (.CEL) files were imported into R and subjected to variance stabilisation normalisation using the VSN package in BioConductor[33]. Quality control was performed using the Bioconductor package arrayQualityMetrics[34] with correction for batch variation performed using the Bioconductor package ComBat[35]. Differential expression was conducted using the Bioconductor package limma in R[36].

Samples identified as outliers during QC filtering were excluded, resulting in 44 samples being taken forward for further network analyses (Supplementary Fig. 2).

To generate a transcriptional signature specific to NK8+ cells (Supplementary Data 3), 50 ml of peripheral whole blood was collected from five healthy volunteers, from which PBMC were isolated by centrifugation over Histopaque (Invitrogen). NK8+ (CD3−CD56+CD8+), NK8− (CD3−CD56+CD8−) and CD3+8+ T cells were isolated by flow sorting following staining with anti-CD56 (Clone REA196),

**Fig. 3 NK8 and sensitivity to HLA-G-mediated suppression. A** Representative histograms (left) and scatterplots (right) of inhibitory HLA-G binding (KIR2DL4, ILT2) and activating (NKG2D) NK receptors on NK8$^+$ and NK8$^-$ subsets, *$P = 0.02$, Wilcoxon paired signed rank two-tailed test, $n = 7$ independent biological replicates per group; error bars = mean +/− sem. **B** Representative histograms illustrating NK activation (CD107 expression, x-axis) on the NK subset (NK8$^-$ top, NK8$^+$ bottom) on coculture with target cell lines (K562/Jurkat, columns) in the presence (red/blue) or absence (black) of sHLAG at 0.1 µg/ml. **C** Dot and line plot of CD107 expression (y-axis, MFI ratio vs vehicle control) on coculture of NK8$^+$ (red) or NK8$^-$ (blue) with target cell lines (K562/ Jurkat) in the presence of a titrated range of soluble HLAG (x-axis). $n = 6$ per group, $P = $ 2-way ANOVA. **D** Heatmap showing unsupervised hierarchical clustering of 94 CIS/RRMS cases (GSE32915) by gene expression (red:blue, max:min) of an optimised panel of surrogate markers reflecting the NK8$^+$ signature in PBMC samples (Supplementary Fig. 8). **E** Scatterplot showing bidirectional enrichment (GSEA$p < 0.05$) of optimal NK8$^+$ surrogates in patient subgroups identified in (**D**). **F** Kaplan-Meier plot showing censored relapse-free survival (y-axis, %) of patient subgroups defined in (**D**), $P = $ log rank test. Error bars = mean +/− sem.

anti-CD8a Fab (Clone BW135/80 Fab) and anti-CD3 REA (Clone REA641) antibodies. Cells were sorted into RNAlysis medium (Tempus Blood RNA Tubes, ThermoFisher) and total RNA was extracted from each cell population using an RNeasy mini kit (Qiagen) with quality assessed using an Agilent BioAnalyser 2100 and RNA quantification performed using a NanoDrop ND-1000 spectrophotometer. cDNA libraries were generated with the SMARTer stranded total RNAseq kit (Takara) with RNAsequencing performed on an Illumina HiSeq4000 instrument following the manufacturer's instructions. Sequences were processed, cleaned and aligned using a combination of bbsplit, hisat2 and trimgalore with count data read into R using the subRead package (Bioconductor). Normalisation and transformation were undertaken using edgeR and voom packages for Bioconductor in R with pairwise differential expression analysis (FDR threshold 5%) using edgeR.

Data QC was ensured using the arrayQualitymetrics (microarray) and fastqc, qorts and pcaexplorer (RNAseq) packages for Bioconductor in R.

**WGCNA analysis.** Unsupervised identification of co-correlated genes was undertaken using the Weighted Gene Coexpression Network Analysis Bio-conductor package in R[16]. Normalised, log-transformed expression data were variance filtered using the median absolute deviation and a soft thresholding power was chosen based on the criterion of approximate scale-free topology[37]. Gene networks were constructed and modules identified from the resulting topological overlap matrix with a dissimilarity correlation threshold of 0.01 used to merge module boundaries with a specified minimum module size of $n = 30$. Modules were summarised as a network of modular eigengenes, which were then correlated with a matrix of clinical variables (Supplementary Data 1 and Supplementary Figure 2). Significance of correlation between a given clinical trait and a modular eigengene was assessed using linear regression with Bonferroni adjustment to correct for multiple testing. Modular signatures were compared to clinical laboratory measures, supporting clinical data and the time to the primary and secondary endpoints of the STAyCIS study (Supplementary Fig. 2 and Fig. 1). Overlap of signatures with modules derived from network analysis (Supplementary Fig. 1F) used a proportional representation method defined by the following formula to allow correction for variable module size: (signature genes in module, $n$)/ (genes in module, $n$) $\times 100$. Hierarchical clustering was performed using a Pearson correlation distance metric and average linkage analysis performed using Genepattern[38].

**Enrichment analyses.** Modular signatures were compared against cell and tissue-specific signatures as defined in the Human Gene Atlas (HGA, GSE1133) and Immune Response In Silico (IRIS, GSE22886) datasets. In brief, these datasets have defined cell and/or tissue-specific patterns of gene expression comparing across data from 79 human/61 murine tissues (HGA) and 22 immune cell subsets (IRIS), respectively. Outcome- associated modular signatures were compared to all cell/ tissue types to confirm specificity, with radarplots shown in Fig. 1) include comparison against all tissues with at least one overlapping transcript. Enrichment analyses were performed using Enrichr[17] or fgsea (Bioconductor) with a significance threshold of 5% FDR.

**Functional NK assays.** Primary human NK cells and CD4$^+$ and CD8$^+$ T cells were separated from leucocyte cones obtained from NHS Blood and Transplant (Addenbrooke's Hospital, Cambridge, UK) by centrifugation over ficoll and either positive selection using magnetic beads (MACS, Miltenyi) according to the manufacturer's instructions or by flow cytometric sorting (NK cell subsets CD3$^-$56$^+$ CD8$^{+/-}$, Supplementary Fig. 9), using anti-CD56 (Clone REA196), anti-CD8a Fab (Clone BW135/80 Fab) and anti-CD3 REA (Clone REA641). Purity of separated cell subsets was determined by three-colour flow cytometry and purified T cells were labelled with 10 µM CFSE (Invitrogen) and resuspended in complete RPMI 1640 (Sigma Aldrich) in the presence of 10% FCS. Purified CD4$^+$ and CD8$^+$ T cells (>95%) were then stimulated in sterile, U-bottomed culture plates (Greiner) using microbead particles conjugated to anti-CD2/3/28 antibodies (1:2 bead:cell ratio, Miltenyi) in the presence of IL2 (10 ng/ml, Gibco Life Technologies) for 6 days.

Natural killer cells were cultured in the presence of autologous stimulated CD4$^+$ T cells, CD8$^+$ T cells or target K562 or Jurkat T cells as indicated for 6 days

with or without titrated range (0.1-0.2 µg/ml[39]) of soluble HLA-G (Abcam) before analysis. Flow cytometry analysis of cultured cells comprised panels including Live/ Dead Blue reactive dye (Life Technologies), anti-CD56 (Clone REA196), anti-CD8a Fab (Clone BW135/80 Fab), anti-CD3 REA (Clone REA641), anti-CD107 (Biolegend, Clone H4A3, used following Brefeldin A protein transport blockade, eBiosciences), anti-HLADR (Clone L243), Granzyme B (Biolegend, clone GB11), Perforin (eBioscience, eBioOMAK-D) and anti-HLAG (Biolegend, clone 87 G) as indicated.

For NK coculture with autologous T cells, it was necessary to account for variable simulation bead:target cell ratio occurring due to titration of increasing numbers of NK cells into culture. Consequently, the ratio of NK activation (CD107 MFI) seen with NK8$^+$ cells was compared to that seen on coculture with a comparable number of NK8$^-$ cells (Fig. 3).

**Flow cytometric immunophenotyping.** Heparinised peripheral blood samples were obtained for analysis of lymphocyte populations and subpopulations by flow cytometry. Whole blood was collected in sodium heparin vacutainers (Becton Dickinson) and shipped ambient overnight to the ITN Flow Cytometry Core (Roswell Park Cancer Institute). Using a stain-lyse method, cells from blinded samples were labelled with 5-colour monoclonal antibody panels using anti-human CD8-PE-Cy5, CD57-FITC (clone NK-1), CD56-PE (clone NCA-1) CD14-APC (clone MφP9), plus CD3-PE-Cy7 (clone SK7, all BD Biosciences). Following staining, data were acquired on a FACSCanto flow cytometer (BD Biosciences), and analysed using WinList's™ (http://www.vsh.com) FCOM function[40].

A validated panel of KIR receptors[22] was used to quantify NK cell subset expression of activating and inhibitory receptors for which clone ids and reagent sources are detailed in Supplementary Data 4. Flow analysis was undertaken using an LSR Fortessa (BD) in the NIHR Cambridge BRC flow phenotyping hub (Supplementary Fig. 9).

**Electron microscopy.** Transmission electron microscopy was undertaken on purified, pooled NK cell subsets sorted as above and processed at the Cambridge Advanced Imaging Centre. Lytic granules were counted and cell size quantified in at least 10 images from each of NK8$^+$ and NK8$^-$ subsets from each of three biological replicates (Supplementary Fig. 5).

**Surrogate NK signature marker identification and validation.** Optimal surrogate markers for identification of the NK8$^+$ signature in PBMC-level data were determined using a random forests classification algorithm[41] (Supplementary Fig. 8). The NK8$^+$ signature itself cannot be directly used in a mixed cell population due to the confounding influence of transcripts from other cell types[42,43]. Expression data derived from both MACS-purified CD8$^+$ cells and PBMC were available for a cohort of $n = 47$ patients' anti-neutrophil cytoplasmic antibody associated vasculitis (AAV) following QC and hybridisation to the HsMediante25k custom microarray platform and constituted a training cohort. Normalised, log-transformed expression data were analysed using the ML Interfaces Bioconductor package in R[44]. Using CD8$^+$-level expression data, AAV samples were classified into subgroups showing either high or low expression of the NK8$^+$ signature (as illustrated in Supplementary Fig. 8). Subsequently, PBMC-level data from the same AAV samples were used to identify and rank probes for their ability to discriminate the NK8$^+$-defined subgroups using the variable Importance metric, reflecting the change in accuracy of classification (% change in Gini coefficient, Supplementary Data 5) when that variable is randomly permuted. Optimal probes were then used to identify analogous patient groups in an independent validation cohort of PBMC data from RRMS/CIS patients (GSE15245[29]). For the validation set, data were downloaded from GEO and imported into R using the Bioconductor package GEOquery[45] in R. The validation study represented a prospective collection of 94 PBMC samples taken from CIS/RRMS cases ($n = 32/62$) and followed up to 3.5 years recording clinical relapses (Supplementary Data 7). Microarray data were generated using Hu133A[29].

**Reporting Summary.** Further information on research design is available in the Nature Research Reporting Summary linked to this article.

## Data availability

The microarray data generated during and/or analysed during the current study are available in the GEO repository accession number E-MTAB-9637. Source data are provided with this paper.

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

## Acknowledgements

This work is supported by the National Institute of Health Research (NIHR) Cambridge Biomedical Research Centre (BRC) and funded by a Medical Research Council (UK) programme award (MR/L019027). E.M. was a Wellcome –Beit Research Fellow supported by the Wellcome Trust and Beit Foundation (104064/Z/14/Z). K.G.C.S. was a Lister Prize Fellow. We thank Anna Petrunkina-Harrison and Natalia Savinykh at the NIHR Cambridge Biomedical Research Cell Phenotyping Hub for extensive assistance. The authors also thank their colleagues at the Immune Tolerance Network, and their collaborators who contribute in many capacities to Immune Tolerance Network projects and perspectives. The authors are grateful to the ITN020AI STAyCIS study participants, and thank the clinical site investigators and study coordinators. Research reported in this publication was supported by the Immune Tolerance Network and sponsored by the National Institute of Allergy and Infectious Diseases of the National Institutes of Health under Award Number UM1AI109565. The content is solely the responsibility of the authors and does not necessarily represent the official views of the National Institutes of Health, the NIHR or the Department of Health. The STAyCIS clinical trial was performed as a project of the Immune Tolerance Network (ITN020AI, ITN contract number N01-AI-15416) within a clinical research consortium sponsored by the National Institute of Allergy and Infectious Diseases (NIAID).

## Author contributions

E.F.M. designed the experiments and conducted them with the help of I.C., C.C., G.M. and E.J.C. Analysis was undertaken by E.F.M. with review by K.H., D.S., S.Z., G.N. and K.G.C.S. The STAyCIS trial was conducted and samples were collected by the ITN STAyCIS study group with S.Z. as principal investigator. The manuscript was written by E.F.M. along with I.C., G.M., K.H., D.S., S.Z., G.N. and K.G.C.S.

## Competing interests

The authors declare no competing interests.
