## [Peer Review File · Nature Communications]

Reviewers' comments:

Reviewer #1 (Remarks to the Author):

Note: It was not possible to find any methods uploaded for this study, so this review was performed without knowing what these were.

Study findings:

This study uses unsupervised network transcriptomic analysis of sorted CD8+ cells from the peripheral blood of patients with CIS with subsequent sampling and analysis of those patients who then experienced a relapse. They reveal a subset of NK cells (CD3-, CD56+) within this CD8 population, which correlate with a positive clinical outcome i.e patients were less likely to relapse, and hypothesise, using in vitro assays, that the mechanism of action is through their superior ability to suppress autologous CD4 T cells compared to their CD8- counterparts.

Comments:

Despite identifying a potentially important subset of NK cells in this study, it is unclear why CD8 was chosen as the primary target for sorting PBMCs – a couple of lines of explanation would be helpful to lead the reader into the study.

CD8 is also expressed on human monocytes and therefore these cells could form one of the modules. The associated text should consider their presence to some extent.

Figure 1C identifies a strong enrichment for NK cell-specific transcripts - such as? There is a lack of detail given about the phenotype of these cells. How was this conclusion made? A supplementary table of this data or a different type of figure to demonstrate this would be helpful due to the current lack of granularity in these findings.

In vitro assays show that CD8+ NK cells have a higher suppressive effect on proliferating CD4 T cells than CD8- NK cells – what stimulation was given here? (appreciate there are no methods shown) but this should be included in figure legend.

HLA-G-associated inhibition of CD8- NK cells to explain their inferior suppressive effects was hypothesised based on comparative inhibitory KIR profiles. This was tested in a separate assay, which did not include CD4 T cells. Could this be proven with more confidence using HLA-G blockade in the same CD8/CD4 assay to exclude other contributing factors?

Figure 3B – what is the x-axis measuring?

Overall, the findings in two separate patient cohorts suggest that measuring the proportion of CD8+ NK cells at disease onset may provide a supportive measure in which to stratify patients into appropriate risk/severity groups. NK cells are clearly performing a role in disease based on other studies, including those in relation to treatment response (eg. their expansion in response to DMF. Marastoni et al. Front Imm. 2019), but their exact function, which is likely to differ according to subset, remains less well defined. This study provides some insight into this, however the data that has been obtained could be better showcased and could provide a more appreciable level of resolution to be used to improve future phenotyping assays, which may become part of a future clinical assessment portfolio.

Reviewer #2 (Remarks to the Author):

McKinney et al present in this well written manuscript the role of CD8+ NK cells on MS clinical

course, detailing their immunoregulatory role on activated CD4+ T cells. Authors used a transcriptomic analysis of CD8+ lymphocytes correlating findings of eigengenes with prospective data obtained from a Phase II clinical trial (STATyCIS, atorvastatine vs. placebo) in a population of RRMS patients. Four gene modules were found to be associated with clinical data, and one of them were enriched for NK cell specific transcripts, which reflected an expanded population associated with reduced risk of relapse after a first demyelinating event.

Afterwards, and following a similar approach used by other authors (Bielekova B, et al. PNAS 2006; 103: 5941-5946; Nielsen N, et al. PLoS One 2012; 7: e31959; Gross CG, et al, PNAS 2016; 113: E2973-82; Laroni A, et al. J Autoimmun 2016; 72: 8-18), NK cells were studied in functional experiments as potential immunoregulators of activated T lymphocytes (CD4+ and CD8+), showing a prominent role of NK8+ cells to control autologous CD4+ T cells. This finding was related to a lower expression of inhibitory receptors on NK8+ cells, which were not counterbalanced by HLA-G expression on target cells.

CD8+ NK cells are known to have a higher cytolytic capability as compared to CD8- NK cells (Addison EG, et al. Immunology 2005; 116: 354-361). By the other hand, NK cell immunoregulatory killing of CD4+ T cells is a well-documented property of these lymphocytes (Waggoner SN, et al. Nature 2011; 481: 394-8). Thus, the results of the study are a straightforward reinforcement of previous studies applied to the field of NK cells in MS. The manuscript is original in its approach to the study of the involvement of NK cells in MS using an unbiased analysis of CD8+ lymphocytes, providing a potential tool to identify those patients with a higher risk of relapses, which could be a valuable biomarker in a disease characterized by a huge heterogeneity and unpredictable clinical course.

Some questions are raised after reviewing the manuscript:

MAJOR COMMENTS:

1.- The study methodology is based on unsupervised network transcriptomics evaluating peripheral CD8+ leucocytes. Whereas the approach is unbiased, the decision to study CD8+ is not. Authors should clearly justify why they have decided to analyze CD8+ T cells, instead of other lymphocytes also involved in MS (e.g. CD4+ T cells, B cells).

2.- Samples were obtained from the STATyCIS phase II clinical trial. The primary endpoint (PEP) of the study was a combination of clinical and radiological outcomes: 1) the development of a clinical relapse within 12 months of enrollment, and 2) the presence of ≥ 3 T2 lesions in serial MRIs. STATyCIS showed that atorvastatin significantly decreased new MRI T2 lesions, but failed to achieve any significant results on relapses, thus not meeting primary endpoint. Whereas in the manuscript (Figure 1A), it is detached time to PEP and time to clinical flare, no information is provided concerning radiological data (except for some variables in Supplementary Table 1). In this regard, it is not clearly stated in the manuscript (i.e. Fig 1D) if the PEP variable used was the combined clinical and MRI endpoint of the STATyCIS or only clinical criteria of time to clinical flare.

Authors should clarify whether radiological variables have an influence in the main results of the study (i.e., association with CD8+ NK+ cells). If this is not the case, PEP as primary endpoint (combined clinical and radiological) should be change for time for clinical flare.

3.- Some important clinical information is missing in the manuscript, both for the STATyCYS trial participants and for the replication cohort.

- Where all patients (n=81) of the STATyCIS analyzed? For the sake of clarity, the manuscript should also clarify that MS patients were untreated at the time of analysis (as specified in the methodology of the STATyCIS clinical trial).

- Were basal demographic and clinical characteristics of patients related with differences in NK8+ cells? In this regard, clinical information concerning the independent cohort of 94 RRMS cases (GSE32915) should also be evaluated. This independent cohort was obtained from a previous study (Reference 27, Gurevich M, et al, BMC Med Genomics 2009; 2:46). Authors have to prove that there were not any clinical or demographic confounders (e.g., interferon-beta or other immunomodulatory therapies) associated with low NK8+ (Figure 2E). This fact is relevant since 32 of the patients (34%) in this cohort initiated immunomodulatory treatment during the follow-up, which could promote a reduction of NK cells (Martínez-Rodríguez JE, et al, Clin Immunol 2010; 137: 41-50).

- In the manuscript it is specified a period of 210 days from CIS onset to baseline visit (Supplementary Figure 1A), whereas in the original STATyCYS study, CIS patients were included within 90 days of symptom onset (Waubant E, et al, Neurology 2012; 78: 1171-1178). In this regard, since NK cells are known to decrease in periods of clinical MS activity, it would be relevant to evaluate whether basal levels of NK8+ in patients were indirectly related to the time from last relapse, excluding a possible bias related with regression to the mean in those patients with a lower risk of relapse and higher NK8+ cells.

4.- "A supervised clustering using the 'black' transcripts identified patient subgroups with distinct outcomes..." (lines 72-74).

How were defined the groups NK8low and NK8high? (Fig 2D)

For a translational clinical approach, do they have any correlation with cytometric values in MS patients expressed as percentages of NK cells?

The high NK8+ population identified in the study, was just a reflection of a general reduction in NK cells in relation to clinical activity? or specifically of these population within NK cells?

5.- Differences regarding CD56 dim and CD56bright NK cells: CD56bright NK cells may also express CD8. This fact is clearly illustrated by authors in Suppl.Fig 3B. Thus, the sentence regarding a similar contribution of CD56hi and NK8+ populations to regulation of autologous CD4+ T cells (lines 147-148) is inaccurate.

6.- Expression of CD8 in non-T cell populations: CD8 is expressed in alpha/beta heterodimers in T cells, whereas in NK cells CD8 is formed by alpha/alpha homodimers, which are characterized by a lower mean fluorescence intensity in flow cytometry. Thus, when authors defined CD8 expression in NK8low or NK8high, it can be misunderstood for the pattern of CD8 expression considered "bright" in T cells and "dim" in NK cells. A sentence explaining differences in CD8 expression in T cells and NK cells would avoid misinterpretation.

7.- Using a cytometric profiling, a reduced frequency of CD8dim NK cells in untreated CIS and RRMS patients as compared to controls was previously described (De Jager PL, et al, Brain 2008; 131: 1701-1711). These population was further identified as CD8lowCD56+CD3-CD4- (NK cells). However, this reduction in NK cells was not correlated with further clinical data. McKinney et al provide in this study interesting data correlating this population (NK8+ cells) with relapse risk. However, it would be relevant to confirm that NK8+ are lower in CIS/MS as compared to a control group to evaluate if the low NK8+ population described in the study is a general feature of MS or it is only perceived in those patients with a higher risk of relapses.

8.- KIR2DL4 may have a dual function as inhibitory or activating receptor (reviewed in Campbell KS, Purdy AK. Immunology 2011; 132: 315-325). KIR2DL4 is an unusual KIR receptor which has a long cytoplasmic tail, typical of inhibitory receptors. However, engagement of this receptor may result in enhancement of cytokine secretion. For the sake of precision, lines 106-107 and hypothesis concerning a lower expression of inhibitory receptors on CD8+ NK cells should have to be drawn up accordingly.

9.- Statistical considerations:

- Sample size in Supplementary figure 5 (n=4-7) is too low to draw conclusions. Increasing sample size could allow the detection of possible significant differences for some KIRs (e.g., KIR2DL2, KIR2DS2) and NKG2A expression.

- Flow cytometry quantification of CD4+ and CD8+ T cell proliferation and activation is evaluated in Figure 2 using 2-way ANOVA. This statistical test evaluates the effect of two independent categorical factors on a dependent continuous variable. From the information presented in the manuscript, it can be deduced that the continuous variable is the T cell proliferation/activation. However, it is not clear which were the categorical variables introduced in the evaluation (the NK:T cell ratio?).

In this regard, authors showed relevant and clear data regarding the inhibition of proliferation (CFSE) and activation (HLA-DR) of CD4+ T cells by NK8+. Comparable results were not found studying CD8+ T cells, except for a lower expression of HLA-DR when a 4:1 ratio of NK8+ cells were analyzed (Fig 2F, NK8+, 47 vs. NK8-, 84), although values expressed as a ratio relative to paired NK8 cell-enriched culture did not showed any differences. Based on this figure, it could be speculated whether higher ratios of NK8+:CD8+ T may render significant results as found for CD4+ T cells.

- Please specify in Figure 3 the sample size and the groups that were pair-wise compared using Mann-Whitney.

MINOR COMMENTS:

- Supplementary Figure 1B-F shows comparable results studying frozen and fresh samples. Resistance to cryopreservation is a well-known characteristic of T lymphocytes and NK cells. Although authors clearly demonstrated in the study the viability of the frozen samples for the analysis, consider abbreviation of the figure and data provided.

- Please provide titles for supplementary tables. In addition, provide definition of abbreviations used.

- Please correct the term "StayCIS" for "STAYCIS" in legend of supplementary Fig 1 and 2.

- "(A)" is omitted in legend of supplementary Fig 1.

- Arrangement of figures and text in Figure 2 should be revised for the sake of clarity.

- Please specify all x-axis labels in supplementary figure 5.

- The independent cohort of MS patients included was constituted by 94 RRMS and CIS patients (62 and 32, respectively), whereas in the manuscript it is stated that they were all RRMS (line 28) (Gurevich M, et al, BMC Medical Genomics 2009). In addition, in Figure 3 it was mentioned that this cohort was formed by was 92 cases instead of 94.

Reviewer #3 (Remarks to the Author):

In their manuscript 'A subset of regulatory CD8+ Natural Killer cells predict favourable clinical outcome in relapsing multiple sclerosis', the authors attempt to demonstrate that a specific subset of CD8 positive natural killer (NK) cells is associated with a favorable clinical outcome in multiple sclerosis (MS) after a first demyelinating event.

As far as it can be deduced from limited available information, the author's performed a microarray-based transcriptional analysis of frozen peripheral blood mononuclear cells after enriching them for CD8 positive cells. Thereby, they identified that a transcriptional module resembling natural killer cells is specifically associated with the subsequent clinical course of MS. Overall the study identifies one interesting novel concept: identifying a potential predictive parameter for the clinical course of MS. If substantiated, such a predictive parameter would be of major relevance to the field as the course of MS can very poorly predicted at present. However, the authors fail to provide their data in a format that allows the reader to understand or retrace any of their experimental and analytical procedures. The main text is written in a non-scientific way without providing any useful details and it additionally appears as if a supplementary method section was inadvertently omitted from the submission. Also,

In conclusion, the study might identify some interesting findings, but it is impossible to appreciate or even evaluate them. I do not understand the format of the text and the overall inadequate way of presentation. Overall, I recommend rejecting this manuscript.

Major Points

The first part of the results section is not a result section, but lines 34 to 50 basically again introduce the field, similar to the introductory paragraph. This is redundant. If this is supposed to be a result section, it is not written in an acceptable way.

Since no methods section is available, none of the experimental procedures can be evaluated. For example, the following (and many more) clinical and experimental aspects remain unknown: What exactly were primary and secondary endpoints? Were only atorvastatin treated or untreated patients used? How many patients were included in the discovery trial? What were the exact inclusion and exclusion criteria? How many patients and samples blood cells were analyzed at what time points? How exactly was transcriptomics was performed? It can be concluded from the text that this was probably microarray, but how and which? How exactly were CD8 positive cells sorted? What was the exact duration follow-up for each patient?

Similarly, no analytical procedures are described or explained in the manuscript. For example, the following information (and many more details) are missing: How exactly were modules identified and defined? Which genes define the so-called 'black' module? Which genes are the CD8 cell intrinsic eigengenes? Which transcripts exactly were associated with T-cell exhaustion and how were they chosen?

The original discovery cohort was based on a clinical trial. The authors should explain whether this was an investigator initiated trial or whether this was a sponsored research agreement and whether the sponsor had any role in performing or initiating the analyses described in the present manuscript.

It is unclear to me why the authors focused on purified CD8+ leukocytes in the first place. Was this based on some former knowledge? Why not CD4+ and CD19+ as well? Were other subsets of peripheral blood analyzed?

The authors likely MACS sorted CD8+ cells from peripheral blood mononuclear cells and followed patients up for an unknown duration of time (maybe 12 or 18 months). And within these CD8+

positive cells there was an enrichment of a so-called black module. Although it remains unclear what exactly this module comprises, it can be deduced from the text that this module was enriched in NK cell transcripts. However, this seems like a self-fulfilling prophecy. When you sort for cytotoxic cells based on a surface marker then you will – naturally – discover cytotoxicity transcripts such as NK cell transcripts. What is the unexpected here? Can the authors provide something like a negative control? For example, a module that is associated with CD8+ T cells but DOES NOT show association with MS prognosis? In other words: Is the detection of the so-called black module only caused by the sorting procedure? And if not how can this be excluded?

Minor points

In the introductory paragraph the author's state that treatment of RRMS relapses has advanced greatly. This is incorrect. The treatment of relapses has basically remained unchanged for decades. Only treatment options for disease modification have changed considerably.

Why did the authors name their transcriptional modules by color such as black or steel blue? If the black module is strongly enriched for NK cell specific transcripts, why not give them a conclusive name such as NK cell module?

It is impossible to understand from the text how many healthy individuals were used to perform confirmatory RNA sequencing?

In line 100, the authors state that they used a validated flow cytometry panel to quantify NK cell receptors. How was this validated and by what?

In line 118, the author state that cytotoxicity was measured by surface translocation of the granule protein CD107. However, is this really equivalent to cell death? Why were no apoptosis or necroptosis markers used to quantify cell death?

In line 139, the authors state that their findings could be used as a predictive tool to guide aggressive therapy in relapsing-remitting multiple sclerosis. This it is a bold statement. At best the authors might demonstrate that the proportion of CD8+ cells is associated with the proportion of relapse free patients. But whether this is qualified to instruct treatment decisions is unknown and will require substantial additional work.

In line 143, the authors used the term valleys. I do not understand the meaning of this term here.

The abbreviations in Suppl. Fig. 2D and E are not explained.

It is incomprehensible from the legend of Supplementary Figure 3 whether the cells depicted in panel B were MACS sorted or not. If panel shows cells after sorting, why is there a large proportion CD8-CD56+ in the left panel? It would also help to not depict CD3-CD56+ cells, but instead depict all CD3- cells without limiting the pre-gating to CD56+ cells. Then the reader can retrace the gating.

All supplementary flow cytometry figures should have numbers indicating the proportion of cells in each gate.

I do not understand the use of supplementary figure 1B. Quantifying the mean fluorescence intensity of markers in flow cytometry is a quantification of protein abundance and does not show anything about the mRNA expression level and whether it is affected by freezing and thawing.

**Responses to reviewers' comments:**

**Reviewer #1 (Remarks to the Author):**

Note: It was not possible to find any methods uploaded for this study, so this review was performed
without knowing what these were.

**We thank the Reviewer for their comments and apologise that the methods section was inadvertently**
**omitted. This has now been rectified with the section provided in full.**

**Study findings:**

This study uses unsupervised network transcriptomic analysis of sorted CD8+ cells from the peripheral
blood of patients with CIS with subsequent sampling and analysis of those patients who then
experienced a relapse. They reveal a subset of NK cells (CD3-, CD56+) within this CD8 population,
which correlate with a positive clinical outcome i.e patients were less likely to relapse, and hypothesise,
using in vitro assays, that the mechanism of action is through their superior ability to suppress
autologous CD4 T cells compared to their CD8- counterparts.

**Comments:**

Despite identifying a potentially important subset of NK cells in this study, it is unclear why CD8 was
chosen as the primary target for sorting PBMCs – a couple of lines of explanation would be helpful to
lead the reader into the study.

**We thank the Reviewer for highlighting this point, which has also been raised by both other Reviewers**
**and was not made sufficiently clear. CD8+ cells were selected for detailed analysis as we have**
**previously identified an association between transcriptional signatures of CD8+ T cell exhaustion and**
**clinical outcome in multiple autoimmune diseases (*Nature* 523, 612-616 (2015). We chose to enrich**
**CD8+ cells with the prior expectation that a similar association may be present in MS. This was not**
**found to be the case (Supp Fig 2E-F), although there are interesting parallels between the two**
**mechanisms of exhaustion and NK autoregulation (both being important in controlling immunopathology**
**in the context of chronic viral infection). This rationale is now made clear in the revised manuscript (lines**
**19-36,44) with Supplementary data provided to illustrate the lack of association between CD8**
**exhaustion signatures and outcome in MS (Supp Fig 2E, F).**

CD8 is also expressed on human monocytes and therefore these cells could form one of the modules.
The associated text should consider their presence to some extent.

**After CD8+ T cells, NK cells express the next highest level of CD8 amongst haematopoietic cell lineages**
**with monocytes only expressing negligible levels. The relative expression level of CD8 mRNA is now**
**illustrated in Supp Fig 3B with additional data panels provided to confirm low levels of non-T cell non-NK**
**populations in the post-enrichment fractions used for transcriptional profiling (SuppFig 3C). We have**
**also altered the manuscript text to remove the assertion that no other haematopoietic lineages express**
**CD8 (line 75).**

Figure 1C identifies a strong enrichment for NK cell-specific transcripts - such as? There is a lack of
detail given about the phenotype of these cells. How was this conclusion made? A supplementary table
of this data or a different type of figure to demonstrate this would be helpful due to the current lack of
granularity in these findings.

**Additional detail about the enrichment process is now provided in the Methods section with additional**
**clarification in the form of two new data panels in Fig1C. In brief, we have used two independent**
**sources of NK-specific transcripts, namely the Human Gene Atlas (HGA, Su et al. PNAS**
**2004;101(16):6062-7) and the Immune Response In Silico dataset (IRIS, Abbas et al. Genes Imm 2005;**
**6(4):319-31) which are now both referenced in the manuscript. HGA defines cell/tissue specific**
**transcripts across 79 human and 61 mouse tissue/cell types while the IRIS dataset defines immune cell**
**subset-specific expression across 22 distinct immune cell subsets. The modular signature identified was**
**enriched for NK-specific transcripts from each dataset, and not from other cells or tissues included. This**
**is now illustrated more clearly in Fig1C, E and described/referenced in the text (line 71).**

**We further confirmed the source of the outcome-associated signature by isolating NK8+ cells (as they**
**were not themselves represented in the above repositories) and undertaking RNAseq to identify an**
**NK8+ specific signature. We then confirmed bidirectional enrichment of this NK8+ signature in the MS**
**cohort data (Fig1G).**

**A supplementary Table of transcripts comprising the 'black' NK8+ modular signature and the NK**
**enriched transcripts are now also provided (Supplementary Table 2).**

In vitro assays show that CD8+ NK cells have a higher suppressive effect on proliferating CD4 T cells
than CD8- NK cells – what stimulation was given here? (appreciate there are no methods shown) but
this should be included in figure legend.

Our apologies, the methods section should have been provided and is now. The cells were stimulated
using polyclonal activation with anti-CD3/CD28 beads in the presence of non-limiting supplementary IL2
(10ng/ml). This information is now also included in the Figure legend as suggested.

HLA-G-associated inhibition of CD8- NK cells to explain their inferior suppressive effects was
hypothesised based on comparative inhibitory KIR profiles. This was tested in a separate assay, which
did not include CD4 T cells. Could this be proven with more confidence using HLA-G blockade in the
same CD8/CD4 assay to exclude other contributing factors?

We thank the reviewer for this suggestion which is something that we had also considered. We have
used a standardised NK cytotoxicity assay to quantify the response of NK8+ and NK8- to HLAG-
mediated suppression. This includes MHC-deficient K562 cells as targets for NK-mediated cytotoxicity.
Undertaking a comparable assay adding sHLAG to coculture of NK and CD4+/CD8+ T cells (as was
undertaken in Fig 2) is complicated by the expression of HLAG binding receptors on both the NK cells
and CD4/CD8 T cells. Consequently, while the assay can be done, it is not feasible to interpret any
suppressive effect as being due to HLAG acting on the NK subset, rather than on the T cell population
directly.

Figure 3B – what is the x-axis measuring?

Our apologies that this Figure's labelling was not sufficiently clear. The x-axis has now been clearly
labelled and shows surface expression of CD107 on the NK subpopulations, an accepted marker of
induced NK cytotoxicity (J Immunol Methods 2004 Nov;294(1-2):15-22.)

Overall, the findings in two separate patient cohorts suggest that measuring the proportion of CD8+ NK
cells at disease onset may provide a supportive measure in which to stratify patients into appropriate
risk/severity groups. NK cells are clearly performing a role in disease based on other studies, including
those in relation to treatment response (eg. their expansion in response to DMF. Marastoni et al. Front
Imm. 2019), but their exact function, which is likely to differ according to subset, remains less well
defined. This study provides some insight into this, however the data that has been obtained could be
better showcased and could provide a more appreciable level of resolution to be used to improve future
phenotyping assays, which may become part of a future clinical assessment portfolio.

We thank the Reviewer for their supportive comments and constructive suggestions and agree that our
study has the potential to improve early patient stratification and treatment selection.

Reviewer #2 (Remarks to the Author):

McKinney et al present in this well written manuscript the role of CD8+ NK cells on MS clinical course,
detailing their immunoregulatory role on activated CD4+ T cells. Authors used a transcriptomic analysis
of CD8+ lymphocytes correlating findings of eigengenes with prospective data obtained from a Phase II
clinical trial (STATyCIS, atorvastatine vs. placebo) in a population of RRMS patients. Four gene
modules were found to be associated with clinical data, and one of them were enriched for NK cell
specific transcripts, which reflected an expanded population associated with reduced risk of relapse
after a first demyelinating event.

Afterwards, and following a similar approach used by other authors (Bielekova B, et al. PNAS 2006;
103: 5941-5946; Nielsen N, et al. PLoS One 2012; 7: e31959; Gross CG, et al, PNAS 2016; 113:
E2973-82; Laroni A, et al. J Autoimmun 2016; 72: 8-18), NK cells were studied in functional experiments
as potential immunoregulators of activated T lymphocytes (CD4+ and CD8+), showing a prominent role
of NK8+ cells to control autologous CD4+ T cells. This finding was related to a lower expression of
inhibitory receptors on NK8+ cells, which were not counterbalanced by HLA-G expression on target
cells.

CD8+ NK cells are known to have a higher cytolytic capability as compared to CD8- NK cells (Addison
EG, et al. Immunology 2005; 116: 354-361). By the other hand, NK cell immunoregulatory killing of
CD4+ T cells is a well-documented property of these lymphocytes (Waggoner SN, et al. Nature 2011;
481: 394-8). Thus, the results of the study are a straightforward reinforcement of previous studies
applied to the field of NK cells in MS. The manuscript is original in its approach to the study of the

involvement of NK cells in MS using an unbiased analysis of CD8+ lymphocytes, providing a potential
tool to identify those patients with a higher risk of relapses, which could be a valuable biomarker in a
disease characterized by a huge heterogeneity and unpredictable clinical course.

We thank the Reviewer for their comments and agree that our observations extend observations of NK-
mediated T cell autoregulation in the context of chronic viral immunity into the field of MS.

Some questions are raised after reviewing the manuscript:

MAJOR COMMENTS:

1.- The study methodology is based on unsupervised network transcriptomics evaluating peripheral
CD8+ leucocytes. Whereas the approach is unbiased, the decision to study CD8+ is not. Authors should
clearly justify why they have decided to analyze CD8+ T cells, instead of other lymphocytes also
involved in MS (e.g. CD4+ T cells, B cells).

We very much agree with the Reviewer that our approach is unbiased and that our selection of CD8+
cells as a substrate for profiling warranted better explanation. As indicated in response to Reviewer1
above, we have included clear explanation of this in the text and Supplementary Figure 2E, F.

2.- Samples were obtained from the STATyCIS phase II clinical trial. The primary endpoint (PEP) of the
study was a combination of clinical and radiological outcomes: 1) the development of a clinical relapse
within 12 months of enrollment, and 2) the presence of ≥ 3 T2 lesions in serial MRIs. STATyCIS showed
that atorvastatin significantly decreased new MRI T2 lesions, but failed to achieve any significant results
on relapses, thus not meeting primary endpoint. Whereas in the manuscript (Figure 1A), it is detached
time to PEP and time to clinical flare, no information is provided concerning radiological data (except for
some variables in Supplementary Table 1). In this regard, it is not clearly stated in the manuscript (i.e.
Fig 1D) if the PEP variable used was the combined clinical and MRI endpoint of the STATyCIS or only
clinical criteria of time to clinical flare. Authors should clarify whether radiological variables have an
influence in the main results of the study (i.e., association with CD8+ NK+ cells). If this is not the case,
PEP as primary endpoint (combined clinical and radiological) should be change for time for clinical flare.

We thank the Reviewer for highlighting the importance of the PEP definitions used. In Figure 1A we
have presented the correlation of unsupervised transcriptional modules with both 'PEP' and 'time to
clinical flare'. PEP in this context reflects the combined endpoint of clinical (≥ 1 clinical exacerbation) or
radiological flare (≥ 3 new T2 lesions with or without Gd+ enhancement), while the clinical flare reflects
clinical exacerbations only. The former is the principal primary endpoint for the clinical STATyCIS trial
(Waubant et al Neurology 2012;78(15):1171-8). We have now clarified our use of the term 'PEP' in the
Figure legends (Fig1) and in the Methods section.

3.- Some important clinical information is missing in the manuscript, both for the STATyCIS trial
participants and for the replication cohort.

- Where all patients (n=81) of the STATyCIS analyzed? For the sake of clarity, the manuscript should
also clarify that MS patients were untreated at the time of analysis (as specified in the methodology of
the STATyCIS clinical trial).

We thank the Reviewer for highlighting this important point. We have now clarified that all patients were
untreated at the time of analysis (Supp Fig 1, Methods, introduction line 38) and included the numbers
of patients included in analysis. Samples were not available from all participants and some were
excluded on the basis of processing QC as indicated in Supp Fig2. 56/81 patients had valid samples
collected of which 79% passed through the QC process with 44 included in the final analysis. This has
now been clarified in Supp Figs 1,2 and methods.

- Were basal demographic and clinical characteristics of patients related with differences in NK8+ cells?
In this regard, clinical information concerning the independent cohort of 94 RRMS cases (GSE32915)
should also be evaluated. This independent cohort was obtained from a previous study (Reference 27,
Gurevich M, et al, BMC Med Genomics 2009; 2:46). Authors have to prove that there were not any
clinical or demographic confounders (e.g., interferon-beta or other immunomodulatory therapies)
associated with low NK8+ (Figure 2E). This fact is relevant since 32 of the patients (34%) in this cohort
initiated immunomodulatory treatment during the follow-up, which could promote a reduction of NK cells
(Martínez-Rodríguez JE, et al, Clin Immunol 2010; 137: 41-50).

We thank the Reviewer for raising this point and agree that excluding potentially confounding covariates
is important. We have provided extensive baseline demographic and clinical covariates in
Supplementary Table 1, illustrating the association of each with all transcriptional modules in

Supplementary Figure 2. None of these associations were significant as we have now indicated in
Supplementary Table 1. For the validation cohort we have also now provided demographic data linked
to the NK signature defined subgroups (Supplementary Table 5).

The NK8⁺ signature identified was identified at baseline visit prior to therapy with no difference in the
therapies administered between subgroups identified. Consequently, any impact of immunomodulatory
treatment on NK cell number, proportion or phenotype is not relevant (as the signature was consistently
measured before such therapy was given).

- In the manuscript it is specified a period of 210 days from CIS onset to baseline visit (Supplementary
Figure 1A), whereas in the original STATyCYS study, CIS patients were included within 90 days of
symptom onset (Waubant E, et al, Neurology 2012; 78: 1171-1178). In this regard, since NK cells are
known to decrease in periods of clinical MS activity, it would be relevant to evaluate whether basal
levels of NK8⁺ in patients were indirectly related to the time from last relapse, excluding a possible bias
related with regression to the mean in those patients with a lower risk of relapse and higher NK8⁺ cells.
We have now provided in (Supplementary Table 1 and in Supp Fig 2D) evidence showing a lack of
association between the NK8⁺ signature and clinical traits associated with the baseline demyelinating
episode that resulted in study enrolment. This includes the time from CIS event to study enrolment.
We have also corrected the description of the screening phase duration from 210 days to 90 days
(methods and SuppFig1).

4.- "A supervised clustering using the 'black' transcripts identified patient subgroups with distinct
outcomes..." (lines 72-74).

How were defined the groups NK8^{low} and NK8^{high}? (Fig 2D)

(We presume the Reviewer is referring to Fig1D)

The Reviewer correctly points out that supervised hierarchical clustering can identify patient subgroups
but that it is also necessary to identify which group reflects NK8^{hi} or NK8^{lo}-specific transcripts. For this
we isolated NK8⁺ cells from healthy donors, generating our own transcriptional signature of NK8⁺/NK8⁻
cell subsets. We next undertook reciprocal gene set enrichment analysis, confirming that NK8⁺ specific
transcripts were significantly enriched in the NK8^{hi} group and vice versa (as indicated in Fig 1G).

For a translational clinical approach, do they have any correlation with cytometric values in MS patients
expressed as percentages of NK cells?

We agree that the ability to use %NK cells as a surrogate of the NK8⁺ signature could facilitate
translation and detection of patients at higher risk of relapse. However, while the NK8 signature is
associated with clinical outcome (Fig1A) and in turn with an NK8⁺ population it does not directly follow
that the % of NK8⁺ cells will be strongly associated with outcome. This likely reflects further phenotypic
differences in the NK8 population that are captured by the transcriptional signature but not reflected in
quantification by a smaller number of surface phenotypic markers and is consistent with previous gene
expression associations in other diseases (e.g. *Nature* 523, 612-616 (2015), *J Clin Invest.*
2011;121(10):4170-4179, *Nat.Med* 2010 May;16(5):586-91, *Journal for Immunotherapy of Cancer* 5(18)
(2017), *Bioinformatics*, Volume 35, Issue 14, July 2019, Pages i436–i445).

We now show that % NK cells and %NK8⁺ cells are not significantly associated with the combined PEP
and consequently are not likely to be useful in prediction of disease outcome directly (Supplementary
Fig 4A,B).

The high NK8⁺ population identified in the study, was just a reflection of a general reduction in NK cells
in relation to clinical activity? or specifically of these population within NK cells?

The association between the 'black' NK8⁺ eigengene and NK8⁺ cells is much stronger than that seen
with the % combined NK cells, although both are statistically significant and are now provided in a new
Supplementary Figure (Supp Fig 4C, D).

5.- Differences regarding CD56 dim and CD56bright NK cells: CD56bright NK cells may also express
CD8. This fact is clearly illustrated by authors in Suppl.Fig 3B. Thus, the sentence regarding a similar
contribution of CD56hi and NK8⁺ populations to regulation of autologous CD4⁺ T cells (lines 147-148) is
inaccurate.

We thank the Reviewer for their comments and agree that a minority of CD56^{hi} NK may also be CD8⁺ as
illustrated in (now) SuppFig3C and we have modified the statement to reflect this (lines 150-151).

6.- Expression of CD8 in non-T cell populations: CD8 is expressed in alpha/beta heterodimers in T cells,
whereas in NK cells CD8 is formed by alpha/alpha homodimers, which are characterized by a lower
mean fluorescence intensity in flow cytometry. Thus, when authors defined CD8 expression in NK8^{low}

or NK8high, it can be misunderstood for the pattern of CD8 expression considered “bright” in T cells and
“dim” in NK cells. A sentence explaining differences in CD8 expression in T cells and NK cells would
avoid misinterpretation.

We agree with the Reviewer that NK cells overall expression lower levels of CD8 than CD3+CD8+ T cells
and that the NK8+ population is a CD8+ subset with lower expression levels than their CD8+ T cell
counterparts, but higher than the CD8- NK subset. We have now clarified this in the text as requested
(line 75) and provide additional data in Supplementary Fig 3A and B illustrating protein expression of
major immune subset markers in the enriched fraction used for transcriptomics alongside mRNA
expression levels of CD8 in a range of immune cell subsets.

7.- Using a cytometric profiling, a reduced frequency of CD8dim NK cells in untreated CIS and RRMS
patients as compared to controls was previously described (De Jager PL, et al, Brain 2008; 131: 1701-
1711). These population was further identified as CD8lowCD56+CD3-CD4- (NK cells). However, this
reduction in NK cells was not correlated with further clinical data. McKinney et al provide in this study
interesting data correlating this population (NK8+ cells) with relapse risk. However, it would be relevant
to confirm that NK8+ are lower in CIS/MS as compared to a control group to evaluate if the low NK8+
population described in the study is a general feature of MS or it is only perceived in those patients with
a higher risk of relapses.

We thank the Reviewer for their comments and agree that reduced NK levels have previously been
associated with active MS. The purpose of our study was to associate and characterise, using an
unsupervised analysis, transcriptional signatures associated with clinical outcome in the StayCIS study.
Consequently, confirming previous reports of reduced NK frequency in active MS compared to healthy
individuals fell outside the remit of our analysis. Furthermore, we were unable to include this analysis as
the StayCIS trial did not recruit healthy individuals but only patients following an episode of
demyelination.

8.- KIR2DL4 may have a dual function as inhibitory or activating receptor (reviewed in Campbell KS,
Purdy AK. Immunology 2011; 132: 315-325). KIR2DL4 is an unusual KIR receptor which has a long
cytoplasmic tail, typical of inhibitory receptors. However, engagement of this receptor may result in
enhancement of cytokine secretion. For the sake of precision, lines 106-107 and hypothesis concerning
a lower expression of inhibitory receptors on CD8+ NK cells should have to be draw up accordingly.

We thank the Reviewer for highlighting the complexity of the KIR2DL4 receptor and we have now
amended the text to reflect this (lines 108-109).

9.- Statistical considerations:

- Sample size in Supplementary figure 5 (n=4-7) is too low to draw conclusions. Increasing sample size
could allow the detection of possible significant differences for some KIRs (e.g., KIR2DL2, KIR2DS2)
and NKG2A expression.

We agree with the Reviewer that with a larger sample size, smaller effect sizes may become statistically
significant while the observed differences would become more so. The purpose of the analysis was to
identify the most differentially expressed KIR receptors for further study and consequently expanding the
size of these groups would not impact on our subsequent findings.

- Flow cytometry quantification of CD4+ and CD8+ T cell proliferation and activation is evaluated in
Figure 2 using 2-way ANOVA. This statistical test evaluates the effect of two independent categorical
factors on a dependent continuous variable. From the information presented in the manuscript, it can be
deduced that the continuous variable is the T cell proliferation/activation. However, it is not clear which
were the categorical variables introduced in the evaluation (the NK:T cell ratio?).

We thank the Reviewer for pointing out that we had not made this clear. The categorical variable in
question is indeed the NK:T cell ratio and this has been clarified in the legend for Fig2.

In this regard, authors showed relevant and clear data regarding the inhibition of proliferation (CFSE)
and activation (HLA-DR) of CD4+ T cells by NK8+. Comparable results were not found studying CD8+ T
cells, except for a lower expression of HLA-DR when a 4:1 ratio of NK8+ cells were analyzed (Fig 2F,
NK8+, 47 vs. NK8-, 84), although values expressed as a ratio relative to paired NK8 cell-enriched
culture did not showed any differences. Based on this figure, it could be speculated whether higher
ratios of NK8+:CD8+ T may render significant results as found for CD4+ T cells.

We agree with the Reviewer that higher ratios of NK:T cells (or indeed lower levels of proliferative
stimulus for the CD8 T cells) may have unmasked an inhibitory effect of NK on CD8+ T cells also. We
intentionally selected comparatively low NK:T cell ratios for this experiment (compared to other
published models using ratios >50:1) to reflect likely physiologically relevant cellular interactions. We

have modified the text to speculate that higher ratios may result in CD8 T cell suppression as the
Reviewer suggests (lines 97-99).
- Please specify in Figure 3 the sample size and the groups that were pair-wise compared using Mann-
Whitney.

This has now been included in the legend for Fig3.

MINOR COMMENTS:

- Supplementary Figure 1B-F shows comparable results studying frozen and fresh samples. Resistance
to cryopreservation is a well-known characteristic of T lymphocytes and NK cells. Although authors
clearly demonstrated in the study the viability of the frozen samples for the analysis, consider
abbreviation of the figure and data provided.

We thank the Reviewer for their supportive comments. We are aware that protocols for cryopreservation
and viability rates can vary between groups and we elected to present our data in Supplementary
material to clarify that our results were not influenced by this process.

- Please provide titles for supplementary tables. In addition, provide definition of abbreviations used.
These have now been included.

- Please correct the term “StayCIS” for “STAyCIS” in legend of supplementary Fig 1 and 2.
These have now been replaced as indicated.

- “(A)” is omitted in legend of supplementary Fig 1.
This has now been included.

- Arrangement of figures and text in Figure 2 should be revised for the sake of clarity.
We thank the Reviewer for this suggestion and we have modified the Figure layout to improve clarity.

- Please specify all x-axis labels in supplementary figure 5.
This has now been amended.

- The independent cohort of MS patients included was constituted by 94 RRMS and CIS patients (62
and 32, respectively), whereas in the manuscript it is stated that they were all RRMS (line 28) (Gurevich
349 M, et al, BMC Medical Genomics 2009). In addition, in Figure 3 it was mentioned that this cohort was
350 formed by was 92 cases instead of 94.

We thank the Reviewer for pointing out this error which has now been corrected with an additional
Supplementary Table (Supp Table 5) showing the clinical characteristics of each patient in the validation
cohort.

Reviewer #3 (Remarks to the Author):

In their manuscript ‘A subset of regulatory CD8+ Natural Killer cells predict favourable clinical outcome
in relapsing multiple sclerosis’, the authors attempt to demonstrate that a specific subset of CD8 positive
natural killer (NK) cells is associated with a favorable clinical outcome in multiple sclerosis (MS) after a
first demyelinating event.

As far as it can be deduced from limited available information, the author's performed a microarray-
based transcriptional analysis of frozen peripheral blood mononuclear cells after enriching them for CD8
positive cells. Thereby, they identified that a transcriptional module resembling natural killer cells is
specifically associated with the subsequent clinical course of MS. Overall the study identifies one
interesting novel concept: identifying a potential predictive parameter for the clinical course of MS. If
substantiated, such a predictive parameter would be of major relevance to the field as the course of MS
can very poorly predicted at present. However, the authors fail to provide their data in a format that
allows the reader to understand or retrace any of their experimental and analytical procedures. The
main text is written in a non-scientific way without providing any useful details and it additionally appears
as if a supplementary method section was inadvertently

omitted from the submission. Also,

In conclusion, the study might identify some interesting findings, but it is impossible to appreciate or
even evaluate them. I do not understand the format of the text and the overall inadequate way of
presentation. Overall, I recommend rejecting this manuscript.

**We apologise for frustrating the Reviewer with our data presentation and realise that the inadvertent
omission of a methods section has greatly contributed to this. This has now been provided and the
queries and comments below addressed in turn.**

Major Points

The first part of the results section is not a result section, but lines 34 to 50 basically again introduce the
field, similar to the introductory paragraph. This is redundant. If this is supposed to be a result section, it
is not written in an acceptable way.

**We thank the Reviewer for their suggestions and have now moved text describing the field and
background immunology to the introductory section.**

Since no methods section is available, none of the experimental procedures can be evaluated. For
example, the following (and many more) clinical and experimental aspects remain unknown: What
exactly were primary and secondary endpoints? Were only atorvastatin treated or untreated patients
used? How many patients were included in the discovery trial? What were the exact inclusion and
exclusion criteria? How many patients and samples blood cells were analyzed at what time points? How
exactly was transcriptomics was performed? It can be concluded from the text that this was probably
microarray, but how and which? How exactly were CD8 positive cells sorted? What was the exact
duration follow-up for each patient?

**Clearly the inclusion of our methods section is critical for the Reviewer and reader to appreciate our
experimental approach and this is now provided in detail. The clinical trial (STAyCIS) on which the
current study is based has also been published with full description of trial methods, inclusion/exclusion
criteria and results. These are available in the referenced trial publication (Waubant et al Neurology
2012;78(15):1171-8) and also summarised in the methods section for convenience.**

Similarly, no analytical procedures are described or explained in the manuscript. For example, the
following information (and many more details) are missing: How exactly were modules identified and
defined? Which genes define the so-called 'black' module? Which genes are the CD8 cell intrinsic
eigengenes? Which transcripts exactly were associated with T-cell exhaustion and how were they
chosen?

414

**As above, these details are now provided in the Methods section. Genes comprising the 'black' module
associated with clinical outcome are also now listed in a Supplementary Table (Supp Table 2).
Transcripts associated with CD8+ T cell exhaustion are also defined in the methods section.**

The original discovery cohort was based on a clinical trial. The authors should explain whether this was
an investigator-initiated trial or whether this was a sponsored research agreement and whether the
sponsor had any role in performing or initiating the analyses described in the present manuscript.

**We thank the Reviewer for highlighting this important issue. The STAyCIS trial was an investigator led
clinical trial run by the Immune Tolerance Network with a clinical research consortium sponsored by the
National Institute of Allergy and Infectious Diseases (NIAID). This information is now described in the
methods section and made clear in the manuscript Acknowledgements.**

It is unclear to me why the authors focused on purified CD8+ leukocytes in the first place. Was this
based on some former knowledge? Why not CD4+ and CD19+ as well? Were other subsets of
peripheral blood analyzed?

**We thank the Reviewer for highlighting this point which has also been queried by both other Reviewers
and addressed as above.**

The authors likely MACS sorted CD8+ cells from peripheral blood mononuclear cells and followed
patients up for an unknown duration of time (maybe 12 or 18 months). And within these CD8+ positive
cells there was an enrichment of a so-called black module. Although it remains unclear what exactly this
module comprises, it can be deduced from the text that this module was enriched in NK cell transcripts.
However, this seems like a self-fulfilling prophecy. When you sort for cytotoxic cells based on a surface
marker then you will – naturally - discover cytotoxicity transcripts such as NK cell transcripts. What is the
unexpected here? Can the authors provide something like a negative control? For example, a module
that is associated with CD8+ T cells but DOES NOT show association with MS prognosis? In other

words: Is the detection of the so-called black module only caused by the sorting procedure? And if not
how can this be excluded?

The Reviewer is correct that the experimental process involved MACS enrichment of CD8⁺ cells from
clinical samples taken during the STAYCIS study with clinical sampling, follow-up and collection details
now described fully in the methods section, illustrated in Supp Fig1 and with trial protocol and design
published previously (Waubant et al Neurology 2012;78(15):1171-8).

The enrichment of NK-cell specific transcripts within a population of CD8⁺ cells is not a self-fulfilling
prophecy however and multiple independent lines of evidence are provided in Fig1 to show that this is
the case (with new data now also provided to illustrate this more clearly). Enrichment of NK-specific
transcripts was indeed an unexpected finding as NK cells were not themselves targeted by the
enrichment process (CD8⁺ T cell intrinsic transcripts would be expected and are indeed identified in
other modules). The nature of the NK-specific transcripts has now been clarified as detailed above in
response to Reviewer 1 (lines 49-63, response to Reviewers).

In order to confirm specific enrichment of NK transcripts we compared the outcome associated ('black')
module with two independent databases of cell and tissue specific transcripts (the Human Gene Atlas
and the Immune Response In Silico dataset (IRIS)) and provide both summary statistics for these
enrichments (Fig 1C, E) alongside new data more clearly illustrating the NK specificity of the relevant
genes (Fig1C). As is now made clear (Fig1C, methods), these comparisons include multiple non-NK
populations as controls (ie that are not enriched in the outcome associated module). All other modules
identified in the unsupervised network analysis of the CD8⁺ transcriptome (Fig1A) are CD8⁺ Tcell
modules that are not associated with MS clinical outcome (illustrating the specificity of the NK
association). We further confirm the source of the 'black' module transcripts in Fig1G by sorting CD8⁺
NK cells, generating an NK8⁺ cell specific signature and showing that this is also enriched in the MS
patients with poor outcome associated with the 'black' transcriptional module.

Minor points

In the introductory paragraph the author's state that treatment of RRMS relapses has advanced greatly.
This is incorrect. The treatment of relapses has basically remained unchanged for decades. Only
treatment options for disease modification have changed considerably.

We thank the author for this suggestion and have modified the introductory text accordingly (lines 14,15)

Why did the authors name their transcriptional modules by color such as black or steel blue? If the black
module is strongly enriched for NK cell specific transcripts, why not give them a conclusive name such
as NK cell module?

We have used unsupervised analysis of the whole CD8⁺ cell transcriptome to identify groups of co-
correlated genes that are associated with clinical traits in MS. This approach, followed by subsequent
interrogation of genes comprising outcome-associated modules, allows for discovery of previously
unsuspected associations. The use of colour-coded labels for such modules is an established approach
(widely applied in the context of weighted network analysis, for examples see: Nature Communications
volume 5, Article number: 3231 (2014), Nature. 2015 Jul 30;523(7562):612-6, BMC Bioinformatics
2008;9:559.

The use of such colour coding appropriately highlights the unsupervised nature of the modules' initial
discovery. We did not want to label the 'black' module as an 'NK module' before we had definitively
demonstrated it to be one (as we do in Fig1).

It is impossible to understand from the text how many healthy individuals were used to perform
confirmatory RNA sequencing?

We thank the Reviewer for highlighting this point which is described in the (previously omitted) methods
section. We have also, for clarity, included the numbers (10 samples from 5 biological replicates) in the
Figure legend for Fig1 where the data is presented.

In line 100, the authors state that they used a validated flow cytometry panel to quantify NK cell
receptors. How was this validated and by what?

The flow cytometry panel used has been previously validated by testing against KIR deficient cell lines
stably transfected with individual KIR genes. The validation is referenced in the manuscript (Ref#22)
with additional provision of the staining panels, including antibody clone IDs, in Supplementary Table 3.

In line 118, the author state that cytotoxicity was measured by surface translocation of the granule
protein CD107. However, is this really equivalent to cell death? Why were no apoptosis or necroptosis
markers used to quantify cell death?

The surface translocation of CD107 is an established method for quantitating the functional activation of
NK cells with surface expression levels shown to reflect both cytokine secretion and NK-mediated target

cell lysis (J Immunol Methods 2004 Nov;294(1-2):15-22.). Surface translocation is quantified on a
population of live NK cells with dead and dying cells backgated. The assay is now described in the
(previously omitted) methods section with reference of the background assay details.

In line 139, the authors state that their findings could be used as a predictive tool to guide aggressive
therapy in relapsing-remitting multiple sclerosis. This it is a bold statement. At best the authors might
demonstrate that the proportion of CD8+ cells is associated with the proportion of relapse free patients.
But whether this is qualified to instruct treatment decisions is unknown and will require substantial
additional work.
We agree with the Reviewer that further work will be required before the observations described can be
used to instruct treatment decisions, although they do create the potential for this to happen. We have
introduced appropriate caveats to make this clear in the manuscript (lines 155-6) and have now clarified
with new data (in Supplementary Fig 4) that simply using the %NK8+ cells as quantified by flow
cytometry is not a sufficiently robust means of achieving accurate patient stratification.

In line 143, the authors used the term valleys. I do not understand the meaning of this term here.
We apologise for the confusing terminology and have modified this to 'periods' (line 146).

The abbreviations in Suppl. Fig. 2D and E are not explained.
This has now been amended in the Figure legend.

It is incomprehensible from the legend of Supplementary Figure 3 whether the cells depicted in panel B
were MACS sorted or not. If panel shows cells after sorting, why is there a large proportion CD8-CD56+
in the left panel? It would also help to not depict CD3-CD56+ cells, but instead depict all CD3- cells
without limiting the pre-gating to CD56+ cells. Then the reader can retrace the gating.
We have now clarified in the Figure legend (and in the methods section) how the MACS enrichment
process was undertaken and have included additional scatterplots (Supp Fig 2A) to illustrate more
clearly the phenotype of the CD3- 'contaminant' population as being CD56+CD8^{int} NK cells. The new
scatterplots shown are not limited to CD56+ cells but show biaxial gating of the enriched fraction of cells
by characteristic surface markers the major immune cell subsets, thereby giving a clear overview of the
populations taken forward for transcriptomics.

All supplementary flow cytometry figures should have numbers indicating the proportion of cells in each
gate.
These have now been included.

I do not understand the use of supplementary figure 1B. Quantifying the mean fluorescence intensity of
markers in flow cytometry is a quantification of protein abundance and does not show anything about
the mRNA expression level and whether it is affected by freezing and thawing.
We thank the Reviewer for their comments on this point. During the freeze/thaw cycle it is feasible for
there to be differential cell survival with some cell types or subsets being selectively depleted, for
example through differential sensitivity to the freeze/thaw process. Such non-random losses would not
register in a QC assessment of the retained populations, but nor would the retained material be directly
representative of the original fresh sample. Consequently, we felt it was important to demonstrate that
there was strong correlation between the phenotypic composition of fresh and frozen samples, which is
illustrated in Supp Fig1B.

REVIEWER COMMENTS

Reviewer #1 (Remarks to the Author):

The authors have largely addressed the points raised during the initial review of their study. Having now reviewed this again with the inclusion of the methods there are a few queries that remain.

It is very difficult to follow the study design regarding the timeline of events. In the results section, there is a primary and secondary endpoint. In the methods section there is a primary combined end point. In figure 1a, there is a primary end point, time to clinical flare, event leading to primary end point and phase of event. It would be helpful if the nomenclature was the same for all 3 sections, with a definition of what these are, making it clear how the NK8+ population relates to them.

Minor points:

1. The opening sentence to the results reads rather awkwardly and begins with what is not required before stating what was done. It makes the start of the results difficult to follow from the outset.
2. In the methods section, it reads, 'Cells were sorted into RNAlysis medium (XX)'.
3. Supplementary figure 3B – the text is unreadable.

This study supports the notion that better patient stratification at disease onset will enable more aggressive treatment options to be given to patients who are considered at higher risk of relapse and thus reduce disease severity in the longer term. These results contribute to how we may hope to achieve this, relying on a relatively straight forward additional analysis to routine blood screening tests that can be measured at disease onset.

Reviewer #2 (Remarks to the Author):

After revision of the manuscript, the authors have properly answered the questions raised. The methodology is sound and clearly stated in the paper. Results are interesting, although sometimes not easy to follow, but with a potential translational approach in the MS field. My only concern is regarding a minor point about potential associations of NK8+ populations with modifications of the global NK cell compartment vs. specific NK8+ cells in relation to clinical activity, since data provided by the authors in the Responses to Reviewers do not seem to correspond to the mentioned figure (Supplementary Figure 4 C and D, the latter not included within the manuscript). I have nothing more to add.

Reviewer #3 (Remarks to the Author):

In their revised manuscript 'A subset of regulatory CD8+ Natural Killer cells predict favourable clinical outcome in relapsing multiple sclerosis', the authors demonstrate that a transcriptional pattern expressed by CD8 positive natural killer (NK) cells among peripheral blood mononuclear cells is associated with a favorable clinical outcome in multiple sclerosis (MS) after a first demyelinating event.

In this revised version of the manuscript, the previously omitted supplementary methods section is now included and now adequately evaluating the manuscript is possible.

The authors first performed a microarray-based transcriptional analysis of CD8+ cells enriched from frozen peripheral blood mononuclear cells derived from a clinical trial of patients with a first

demyelinating event. In this dataset, they identified that a transcriptional module resembling natural killer cells that is specifically expressed in patients with a favorable clinical course of MS. They find that features of this module confer NK cells with enhanced suppressive capacity over autologous CD4 T cells through the invariant HLA-G molecule and confirm the predictive potential in a confirmatory patient cohort. Overall, the most interesting and relevant concept introduced by the study is the identification of a predictive biomarker for the subsequent clinical course of MS. Such a predictive parameter would be of major relevance to the field, since the course of MS cannot be well predicted at present.

In conclusion, the study has improved substantially from its previous version, identifies an interesting core finding, and would have relevance to the field. However, several aspects need improvement that could not be anticipated previously due to the lack of sufficient methodical information.

Major Points

The main finding is the potential predictive potential of the NK8 module. How does that perform when compared against known clinical and MRI predictors like for example lesion load, recovery from first relapse, multifocal and spinal manifestation vs. optic neuritis only? Can the authors statistically quantify the predictive potential of the NK8 module for this comparison?

If the current manuscript is correct, then I could start composing a qPCR panel to quantify the top NK8 surrogates in PBMCs collected from my MS patients now. And thereby predict the likely clinical course of these patients. But how would I compose that panel? I cannot find a Suppl. Table that lists these optimal surrogates. Why not show at least some of them in Fig. 3D? Can the optimal surrogates be further prioritized? It seems from Suppl. Fig. 8C that there is some gradient in the importance of these optimal surrogates. Which transcripts fare best? How few surrogates would suffice to predict?

I cannot find an abstract in the manuscript file.

I consider Fig. 1B as the key plot of the manuscript. How would the plot look if it were separated into two plots by atorvastatin treatment vs. placebo? Related to this: is it correct that all CD8+ samples were collected before treatment? Is it then also correct that the atorvastatin treatment cannot have affected the expression of the module? This should be mentioned explicitly

I do not understand why the confirmation cohort in Fig. 3E is depicted as a Kaplan-Meier curve, but the main cohort as volcano plot in Fig. 1B. I think both plots for both cohorts would be interesting.

Some basic clinical details of the cANCA associated vasculitis patients used as negative controls should be provided; eg average age, disease duration, sex, CNS involvement yes/no, treatments, etc.

Minor points

In lines 14-15 the author's state that treatment options of RRMS relapses have expanded. The authors have changed the sentence, but I must insist on this: Relapses themselves have been treated with steroid and plasmapheresis / immune-adsorption for decades. Nothing has changed about this. Only the treatment of relapsing-remitting MS (i.e. preventing relapses, not treating relapses) has expanded dramatically. There is a difference.

The paper by Gross et al. 2016 PNAS should be cited when discussing the function of NK cells in repressing CD4 T cells on pages 1 and 4.

I cannot find a definition of the abbreviations WGCNA (line 58), MSE, and AAV.

Fig 1B: label of the x-axis should probably be: correlation with 'time to' PEP. Otherwise what is the unit of the PEP here?

Legend of Fig. 1F refers to a radar plot on the left, but the plot is missing from the figure.

Fig. 1D: the heatmap probably depicts genes vs. patients, but it should be defined that columns refer to patients.

There is still an 'XX' as a placeholder on the first page of the methods.

In Fig. 3, panels A-C are less important than panels D-F and should be reduced in relative size.

Suppl. Tab. 4 is not referenced anywhere in the text. I cannot find Suppl. Tab. 2 in the uploaded files. This is probably a numbering issue.

Legend of Suppl Fig. 8 refers to a Fig. 4, that does not exist.

REVIEWER COMMENTS

Reviewer #1 (Remarks to the Author):

The authors have largely addressed the points raised during the initial review of their study. Having now reviewed this again with the inclusion of the methods there are a few queries that remain.

It is very difficult to follow the study design regarding the timeline of events. In the results section, there is a primary and secondary endpoint. In the methods section there is a primary combined end point. In figure 1a, there is a primary end point, time to clinical flare, event leading to primary end point and phase of event. It would be helpful if the nomenclature was the same for all 3 sections, with a definition of what these are, making it clear how the NK8+ population relates to them.

We thank the Reviewer for highlighting this point. We agree the terminology could be clearer and have altered this (Figure 1, manuscript lines 76-77) to be both internally consistent and consistent with the STAyCIS trial also, which defined the primary endpoint as either one clinical relapse or at least 3 new T2 lesions within 12 months. The definitions of features included in Fig1A have also been further clarified in the figure legend alongside description in the manuscript (lines 76-77).

Minor points:

1. The opening sentence to the results reads rather awkwardly and begins with what is not required before stating what was done. It makes the start of the results difficult to follow from the outset.

Thank you – the opening statement has been modified to make the point clearer and to improve the flow.

lines 66-68: “Unsupervised analyses of high throughput ‘omic’ data facilitate the discovery of previously unsuspected associations as they are, by definition, not founded on existing knowledge.”

2. In the methods section, it reads, ‘Cells were sorted into RNAlysis medium (XX)’.

Thank you – this omission has been corrected.

Methods, line 43: ‘Cells were sorted into RNAlysis medium (Tempus Blood RNA tubes, ThermoFisher)’

3. Supplementary figure 3B – the text is unreadable.

My apologies – the text format has now been altered to make this clear (Supplementary Figure 3B).

This study supports the notion that better patient stratification at disease onset will enable more aggressive treatment options to be given to patients who are considered at higher risk of relapse and thus reduce disease severity in the longer term. These results contribute to how we may hope to achieve this, relying on a relatively straightforward additional analysis to routine blood screening tests that can be measured at disease onset.

Reviewer #2 (Remarks to the Author):

After revision of the manuscript, the authors have properly answered the questions raised. The methodology is sound and clearly stated in the paper. Results are interesting, although sometimes not easy to follow, but with a potential translational approach in the MS field. My only concern is regarding a minor point about potential associations of NK8+ populations with modifications of the global NK cell compartment vs. specific NK8+ cells in relation to clinical activity, since data provided by the authors in the Responses to Reviewers do not seem to correspond to the mentioned figure (Supplementary Figure 4 C and D, the latter not included within the manuscript). I have nothing more to add.

We thank the Reviewer for their careful review of our work and apologise for the mislabelling in the text of the ‘Response to Reviewers’. This should have referred the Reviewer to Supplementary Figure 3D-G and was appropriately referenced in the manuscript text (now line 100).

Reviewer #3 (Remarks to the Author):

In their revised manuscript ‘A subset of regulatory CD8+ Natural Killer cells predict favourable clinical outcome in relapsing multiple sclerosis’, the authors demonstrate that a transcriptional pattern expressed by CD8 positive natural killer (NK) cells among peripheral blood mononuclear cells is associated with a favourable clinical outcome in multiple sclerosis (MS) after a first demyelinating event.

In this revised version of the manuscript, the previously omitted supplementary methods section is now included and now adequately evaluating the manuscript is possible.

The authors first performed a microarray-based transcriptional analysis of CD8+ cells enriched from frozen peripheral blood mononuclear cells derived from a clinical trial of patients with a first demyelinating event. In this dataset, they identified that a transcriptional module resembling natural killer cells that is specifically expressed in patients with a favourable clinical course of MS. They find that features of this module confer NK cells with enhanced suppressive capacity over autologous CD4 T cells through the invariant HLA-G molecule and confirm the predictive potential in a confirmatory patient cohort. Overall, the most interesting and relevant concept introduced by the study is the identification of a predictive biomarker for the subsequent clinical course of MS. Such a predictive parameter would be of major relevance to the field, since the course of MS cannot be well predicted at present.

In conclusion, the study has improved substantially from its previous version, identifies an interesting core finding, and would have relevance to the field. However, several aspects need improvement that could not be anticipated previously due to the lack of sufficient methodical information.

Major Points

The main finding is the potential predictive potential of the NK8 module. How does that perform when compared against known clinical and MRI predictors like for example lesion load, recovery from first relapse, multifocal and spinal manifestation vs. optic neuritis only? Can the authors statistically quantify the predictive potential of the NK8 module for this comparison?

We thank the Reviewer for their further consideration of the manuscript.

Known clinical and MRI predictors are listed along with extensive clinical metadata in Supplementary Table 1 and Supplementary Figure 2, illustrating that the NK signature identified does not correlate with them, indicating that it provides independent information.

However, as the Reviewer indicates, this doesn't mean that additional information provided by other baseline clinical covariates may not improve predictive power in combination with the NK signature. To set the NK signature in context of the predictive potential of additional clinical covariates we have now incorporated 14 baseline measures (reflecting the clinical, functional and radiological impact of the presenting demyelination episode alongside the NK eigenvector) into a penalised Cox proportional hazards model, identifying the contribution of each feature to the optimal model, as identified through cross-validation (elastic net model, Fig 1H, manuscript lines 90-94). This indicates that the NK signature makes the strongest contribution to predictive accuracy, although there is an additive contribution from radiological (baseline T1 lesion load) and clinical estimates of disease severity.

This analysis confirms that the signature identified is independent of baseline clinical traits and also that it may be used in concert with them to improve predictive accuracy.

If the current manuscript is correct, then I could start composing a qPCR panel to quantify the top NK8 surrogates in PBMCs collected from my MS patients now. And thereby predict the likely clinical course of these patients. But how would I compose that panel? I cannot find a Suppl. Table that lists these optimal surrogates. Why not show at least some of them in Fig. 3D? Can the optimal surrogates be further prioritized? It seems from Suppl. Fig. 8C that there is some gradient in the importance of these optimal surrogates. Which transcripts fare best? How few surrogates would suffice to predict?

We have now provided a Table (Supp Table 5) providing a list of the genes included in the optimal surrogate prediction. Supplementary Figure 8C shows the relative importance of each transcript in a random forests predictive model and Supplementary Table 5 also includes individual feature importance rankings and metrics.

I cannot find an abstract in the manuscript file.

Thank you – an abstract is now included.

I consider Fig. 1B as the key plot of the manuscript. How would the plot look if it were separated into two plots by atorvastatin treatment vs. placebo?

Thank you – Figure 1B illustrates the association between gene coexpression modules and the primary endpoint of the trial. To put this association in context we provided illustration of the same modules' association with other traits also, including by treatment group in the STAYCIS trial (Fig 1A, 'treatment group' and Supp Figure 2B-D). This demonstrates that no modules showed significant association with

treatment group in the trial (Fig 1A, first column). Treatment group is also now explicitly modelled alongside other baseline clinical traits in the penalised Cox model presented in Fig 1H. This shows that treatment group provides negligible support to the prediction of relapse.

Related to this: is it correct that all CD8+ samples were collected before treatment? Is it then also correct that the atorvastatin treatment cannot have affected the expression of the module? This should be mentioned explicitly.

This is correct and has now been mentioned explicitly in the manuscript (lines 54-55).

I do not understand why the confirmation cohort in Fig. 3E is depicted as a Kaplan-Meier curve, but the main cohort as volcano plot in Fig. 1B. I think both plots for both cohorts would be interesting.

We have chosen to use volcano and Kaplan-Meier plots respectively to illustrate the discovery and validation analyses as each analysis is quite different.

Our discovery approach involves unsupervised network coexpression modelling of CD8+ cells isolated from the discovery cohort. This generates multiple summary eigenvectors (one for each coexpression module) which are then correlated with extensive baseline and outcome clinical traits. A volcano plot is an appropriate means of illustrating the correlation of many such modular eigenvectors against the study endpoint (as shown in Fig 1B), something a Kaplan-Meier plot could not achieve (as it would require as many curves as there are modular signatures).

For the validation cohort, we are testing optimal surrogate markers of the NK signature in PBMC samples from MS patients. The optimal surrogate genes are used to identify binary groups of high and low risk which are then associated with time-to-event. As there are only two groups in question, a Kaplan-Meier plot is the preferred method (a volcano plot would only have one data point).

Some basic clinical details of the cANCA associated vasculitis patients used as negative controls should be provided; eg average age, disease duration, sex, CNS involvement yes/no, treatments, etc.

A table of clinical traits relating to the ANCA-associated vasculitis cohort is now included (Supplementary Table 6). No patients had CNS involvement and all were enrolled with samples taken prior to starting immunosuppressive therapy.

Minor points

In lines 14-15 the author's state that treatment options of RRMS relapses have expanded. The authors have changed the sentence, but I must insist on this: Relapses themselves have been treated with steroid and plasmapheresis / immune-adsorption for decades. Nothing has changed about this. Only the treatment of relapsing-remitting MS (i.e. preventing relapses, not treating relapses) has expanded dramatically. There is a difference.

We thank the Reviewer for highlighting this distinction – we have now modified the text to remove the suggestion that there has been an expansion of treatment options for RRMS (lines 29-30)

The paper by Gross et al. 2016 PNAS should be cited when discussing the function of NK cells in repressing CD4 T cells on pages 1 and 4.

This has now been referenced as suggested.

I cannot find a definition of the abbreviations WGCNA (line 58), MSE, and AAV. These have now been included where used.

Fig 1B: label of the x-axis should probably be: correlation with 'time to' PEP. Otherwise what is the unit of the PEP here?

Thank you – this has been altered as suggested.

Legend of Fig. 1F refers to a radar plot on the left, but the plot is missing from the figure.

Thank you – the legend has been corrected.

Fig. 1D: the heatmap probably depicts genes vs. patients, but it should be defined that columns refer to patients.

This is correct and has been made explicit in the figure legend (Fig1D).

There is still an 'XX' as a placeholder on the first page of the methods.

Thank you. This has been corrected.

In Fig. 3, panels A-C are less important than panels D-F and should be reduced in relative size.
The sizes have been adjusted as suggested.

Suppl. Tab. 4 is not referenced anywhere in the text. I cannot find Suppl. Tab. 2 in the uploaded files.
This is probably a numbering issue.
There are now 7 supplementary tables which are referenced in the appropriate order in the manuscript.

Legend of Suppl Fig. 8 refers to a Fig. 4, that does not exist.
Thank you – the legend has been updated to appropriately refer to Fig 3D.

REVIEWERS' COMMENTS

Reviewer #1 (Remarks to the Author):

The various comments from the referees have now been addressed. However, I would strongly recommend the authors to review their manuscript to make sure that there are no further omissions such like a forgotten abstract etc.

Reviewer #3 (Remarks to the Author):

All my previous concerns have been thoroughly addressed. Especially the new panel in Figure 1H considerably supports the key findings of the manuscript. I recommend accepting the manuscript.